# Red-Listed Ecosystem Status of Interior Wetbelt and Inland Temperate Rainforest of British Columbia, Canada

**Dominick A. DellaSala** [1],*, **James R. Strittholt** [2], **Rebecca Degagne** [2], **Brendan Mackey** [3], **Jeffery R. Werner** [4], **Michelle Connolly** [5], **Darwyn Coxson** [4], **Andrew Couturier** [6] and **Heather Keith** [3]

1    Wild Heritage, a Project of Earth Island Institute, 222 Joseph Drive, Talent, OR 97540, USA
2    Conservation Biology Institute, 136 SW Washington Ave, Suite 202, Corvallis, OR 97333, USA; stritt@consbio.org (J.R.S.); rebecca.degagne@consbio.org (R.D.)
3    Griffith Climate Action Beacon, Gold Coast Campus, Griffith University, Brisbane, QLD 4222, Australia; b.mackey@griffith.edu.au (B.M.); h.keith@griffith.edu.au (H.K.)
4    Ecosystem Science and Management Program, University of Northern British Columbia, 3333 University Way, Prince George, BC V2N 4Z9, Canada; Jeffery.Werner@unbc.ca (J.R.W.); Darwyn.Coxson@unbc.ca (D.C.)
5    Conservation North, 606 Freeman Street, Prince George, BC V2M 2R2, Canada; michelle.connolly@alumni.unbc.ca
6    Birds Canada, P.O. Box 160, 115 Front Road, Port Rowan, ON N0E 1M0, Canada; acouturier@bsc-eoc.org
*    Correspondence: dominick@wild-heritage.org

**Abstract:** The Interior Wetbelt (IWB) of British Columbia, which includes the globally rare Inland Temperate Rainforest (ITR), contains primary forests poorly attributed and neglected in conservation planning. We evaluated the IWB and ITR using four IUCN Red List of Ecosystems Criteria: geographic distribution, environmental degradation (abiotic and biotic factors), and likelihood of ecosystem collapse. Clearcut logging (3.2M ha) represented 57% of all anthropogenic disturbances, reducing potential primary forest by 2.7 million ha (28%) for the IWB and 524,003 ha (39%) for the ITR. Decadal logging rates nearly doubled from 5.3% to 10.2% from 1970s–2000s. Core areas (buffered by 100-m from roads and developments) declined by 70% to 95% for the IWB and ITR, respectively. Vulnerable was assigned to karst, the only abiotic factor assessed, because it was associated with rare plants. For biotic factors, Old-Growth Birds were Vulnerable, Southern Woodland Caribou (*Rangifer tarandus caribou*) habitat and Sensitive Fish were Endangered, and Old-Growth Lichens habitat was Critical. Overall, the IWB was ranked as Endangered and the ITR as Critical with core area collapse possible within 9 to 18 years for the ITR, considered one of the world's most imperiled temperate rainforests.

**Keywords:** British Columbia; inland temperate rainforest; interior wetbelt; endangered; critical; collapse

## 1. Introduction

The Earth's forests are experiencing unprecedented loss from human activity [1] (also see https://www.globalforestwatch.org/ accessed on 10 June 2021). Less than one-third remain in primary forest condition; thus, determining conservation status and ecological importance of primary forests globally and regionally is vital to forestall further loss [2]. Primary forest are naturally regenerated forests of native species, including all tree ages, whose structure and dynamics are dominated by ecological and evolutionary processes, including natural disturbance regimes. No clearly visible indications of human activities occur, and ecological processes are not significantly disrupted in primary forests [2–4].

Various approaches have been developed for assessing the conservation status of imperiled ecosystems like primary forests, including global hotspots [5], WWF Global 200 ecoregions [6], roadless areas [7], intact/wilderness areas [8,9], and Key Biodiversity Areas [10]. While these approaches all provide useful perspectives and highlight important attributes, they are not comprehensive enough for assigning status rankings to our study area—the Interior Wetbelt (IWB) of British Columbia (BC), Canada—a large (~16.46M ha)

mainly forested bioregion containing portions of two WWF ecoregions and several biogeo-climatic subzones under intense development [11]. The IWB has not been identified as a hotspot or Key Biodiversity Area to date, although efforts to identify the latter in Canada are at an early stage. Further, there are benefits in applying an international assessment standard that can be compared with any other region, rather than using different national or sub-national methods. The IUCN method has only recently been finalized with guidelines for applying the RLE criteria [12] and our study area represents a test case.

RLE methodologies have been used to assess a wide array of ecosystem types, including wet temperate forests subject to development [13]. For instance, an analysis of the likelihood of ecosystem collapse using the RLE criteria for the mountain ash (*Eucalyptus regnans*) forests of Victoria, south-eastern Australia, resulted in a critically endangered status ranking and recommendations to reduce logging pressures [13]. Here, we focused mainly on RLE criteria for which we had spatial and temporal data, including A1 (reduction of geographic distribution), C1 (environmental degradation—abiotic), and D1 (environmental degradation—biotic). We also applied a general persistence criterion to the IWB and ITR based on whether ecosystem collapse is imminent using RLE Criterion E [12]. Like the mountain ash forest example, logging pressure in the IWB bioregion has been extensive. Thus, our analysis using the RLE criteria may shed light on conservation importance and status of the IWB.

While BC has recognised the importance of old-growth forests (i.e., generally wetter forests > 250 years), provincial government legislation, regulations and policy do not protect primary forests of any age class, let alone old growth [14,15]. Doing so would help Canada's federal government meet its pledge to protect 25% of lands and waters by 2025 and 30% by 2030 [16]. Notably, Canada (13% protected) and BC (15% province-wide, 10% productive old growth protected [15]) are thus far short of this modest target.

Additionally, the IWB has not been assessed for its conservation status and potential contributions to Canada's commitments. Compared to temperate rainforests on the BC coast (i.e., Great Bear rainforest) that have received most of the provincial conservation attention, IWB primary forests have received scant attention and are widely exploited [11,14,15,17] in the absence of contemporary conservation planning. These interior forests support globally significant biodiversity such as extraordinary lichen richness [18,19], contain substantial carbon stocks [17,20], and represent one of only three inland temperate rainforests (ITR) on Earth (Russian Far East and Southern Siberia are the others [11]).

Our specific objective was to evaluate the conservation status of the interior BC forests by applying an RLE assessment of primary forests at multiple scales (IWB bioregion, ITR subzone, watersheds). At least for the ITR, primary is synonymous with old growth given extremely long intervals between natural disturbance dynamics like wildfires in these wet forests. In general, our analysis required an assessment of what remains, where it occurs, and how fast it is being depleted. We focused only on the BC portion of the bioregion because of the need to assess conservation priorities in relation to Canada's federal government protected areas pledge and regional conservation priorities. Additionally, the regional climate, disturbance ecology, and species assemblages change markedly from north to south, presenting cross-border and data incompatibility challenges beyond the scope of our study. We described the conditions of our study area following the RLE methodology [12] with particular attention to classifications, spatial distributions, characteristic native biota and abiotic factors, ecological processes and interactions, anthropogenic threats, and overall status evaluation criteria. To our knowledge this is the first application of the RLE criteria for any forest type in North America.

## 2. Study Area

### 2.1. Classifications

The 16.46M ha study area is located along the windward slopes of the Columbia and Rocky Mountains within southcentral interior BC and includes three major river

basins (Fraser, Thompson, BC, Canada) (Figure 1). The IWB includes portions of two WWF ecoregions, North Central Rockies Forests and Okanagan Dry Forests [21]; and portions of the Central Rocky Mountain Mesic Lower Montane Forests (Macrogroup M500), Central Rocky Mountain Dry Lower Montane-Foothill Forest (M501), and Rocky Mountain Subalpine-High Montane Forest (M020) [22]. The study area boundary was delineated using a combination of biogeoclimatic zonation [23,24] and watershed mapping. The ITR boundary was created by selecting the very wet, wet, and moist interior cedar (*Thuja plicata*)-hemlock (*Tsuga heterophylla*) (ICH) and Engelmann spruce (*Picea englemanni*)-subalpine fir (*Abies lasiocarpa*) (ESSF) subzones from the Biogeoclimatic Ecosystem Classification Codes and Names [24]. The intersecting watersheds from the Freshwater Atlas [25] were selected along with those needed to fill in and defragment the boundary. We used the northern boundary of the ICH layer as the northern cutoff at ~54.5 degrees and a prominent depression just south of the Peace River for the IWB at ~55.3 degrees (Figure 1).

## 2.2. Forest Types

Approximately 10.8M ha (66%) of the study area is forested with most of this forested area (9.43M ha) classified as Interior Wet Belt (IWB). IWB forests are generally distributed at low to mid (400–1000 m-north, up to 1800 m south) elevations dominated by western redcedar and western hemlock grading into hybrid white (often called interior) spruce (*Picea engelmannii x glauca*) and subalpine fir at higher elevations (i.e., snow forests). Drought-tolerant Douglas-fir (*Pseudotsuga menziesii, var. glauca),* western larch (*Larix occidentalis*), and ponderosa pine (*Pinus ponderosa*) occur to the south and western edge of the study area [26]. The wettest forest types are confined to the wet-cool, very-wet cool, and moist cool biogeoclimatic subzones of Sub-Boreal Engelmann Spruce-Subalpine Fir and the moist, wet, and very wet subzones of Interior Cedar-Hemlock (ICH) subzone [24,25]. These ecosystem types are collectively referred to as ITR [11,17,19,26–28]. The ITR component of the IWB totaled ~1.33M ha (8.1% of the total study area, 14.1% of the IWB forest area) and is mainly distributed above latitude 50° N in disjunct valleys west of the continental divide. Additional areas of ITR occur as isolated pockets in the northern contiguous U.S. above latitude 46° N (not in our study area) [11].

## 2.3. Forest Structure

At its northern limit, IWB forests can be described as snow forests with narrow spires of hybrid spruce and subalpine fir forming uneven aged forests from valley-bottom to treeline. In the southern IWB, forest composition is more dependent on slope and aspect with subalpine fir, hybrid spruce, western hemlock, and western larch transitional to grasslands and dry ponderosa pine at low elevation and on south facing slopes. Western redcedar in the southern IWB is typically limited to sites where topography provides protection from fires, as in protected canyons. Contemporary forest structure and composition within the ITR developed some 6000 to 2000 years BP following retreat of Pleistocene glaciers [27]. Consequently, much of the ITR existed in its present form for but a few generations of the oldest rainforest trees and are known as "antique" forests [27]. Notably, western redcedars can live > 1600 years in these forests, attaining massive size (~60-m tall, up to 5-m dbh) (forest metrics data available at https://databasin.org/datasets/6288d3cb6cf74997b118 d8be38785d00/ accessed on 10 June 2021). Primary old growth ITR is characterized by multi-layered tree canopies, diverse forb and shrub understories, canopy gaps, and large dead and down trees (i.e., classic old growth temperate rainforest) [11,19,26–28].

## 2.4. Abiotic Factors

The IWB consists of distinct physiographic subdivisions of the province. The Rocky Mountains include limestone, quartzite, schist, and slate. The Columbia Mountains include four distinct ranges: sedimentary and metamorphic Cariboo Range; gneissic metamorphic, sedimentary volcanic, and intrusive igneous Monashee Range; sedimentary, metamorphic and volcanic and intrusive igneous Selkirk Range; and sedimentary and metamorphic

Purcell Range [28]. The ITR is restricted to narrow-forested strips on middle and lower toe-slopes in U-shaped valleys where discharge from deep winter snows provide seeps (abundant surface springs) [19,26]. The only abiotic factor evaluated in our study area is karst because it supports rare plant assemblages and has conservation status in British Columbia planning decisions [29].

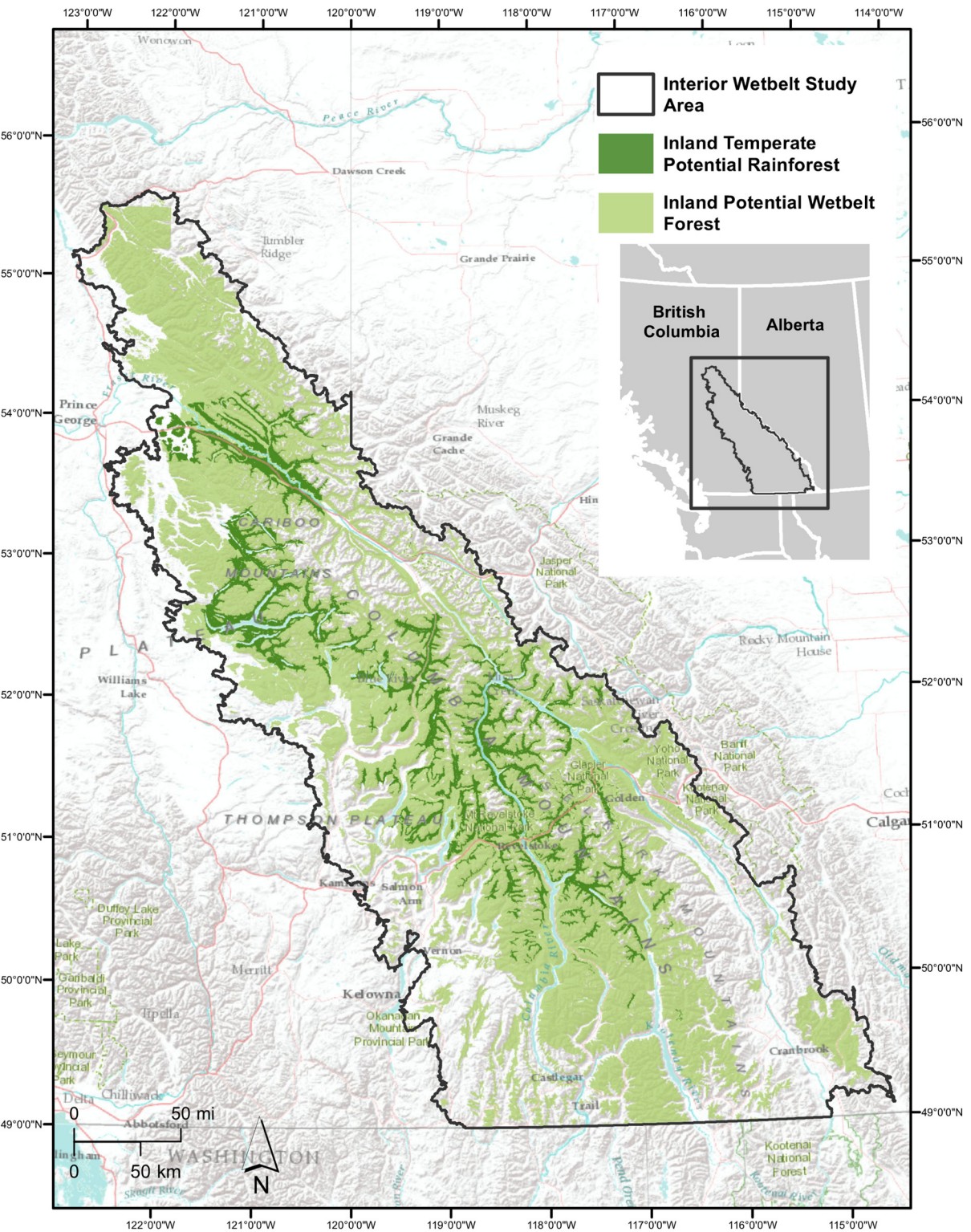

**Figure 1.** Interior Wetbelt (IWB) of southcentral British Columbia showing the Inland Temperate Rainforest (ITR) as a subzone of the wettest portions of the study area.

Regional climate of the IWB is a mixture of maritime and continental influences that vary seasonally and biogeographically. Westerly air masses deliver most of the annual precipitation seasonally during winter and spring, deposited primarily along the windward montane flanks that offer climatically favorable conditions for temperate rainforests [11,26–28]. Weather patterns reverse and become continental in summer, especially southward. Rainforest lichens are restricted to moister microsites southward, such as waterfall sprayzones [30]. Mean temperature during the warmest quarter (June–August) is 13.5 °C (max 17.9 °C, min 10.8 °C but it can get as hot as 30 °C in summer, especially southward [11]. Annual precipitation for the ITR averages ~1000 mm with the driest quarter averaging 152 mm (min 106 mm, max 198 mm) and the wettest quarter averaging 267 mm (min 187 mm, max 458 mm). Overall, the ITR tends to be on the dry side of the global temperate rainforest climatic gradient but on the high side for snowfall [11]. Relatively low annual precipitation levels are compensated by extended snowmelt, groundwater discharge, seeps, and cool night-time and morning temperatures that provide suitable microclimatic conditions for a rich assemblage of rainforest indicator and moisture sensitive lichens and vascular plants.

Fire rotation intervals (landscape scale) within the ITR are on millennial time scales (800 to 1200 years) allowing forests to persist for centuries; however, more frequent large-scale wildfires occur southward in the IWB [31,32]. Periodic epizootic outbreaks and wind/ice storms are principal natural disturbance agents with effects ranging from gap-phase to landscape-scale disturbance dynamics [28].

### 2.5. Biotic Factors

The ITR is perhaps the most species-rich lichen (epiphytic and oceanic cyanolichens) temperate rainforest in the world [18,19] with many taxa only recently described and others yet to be identified [33–35]. Epiphytic lichens have significant (~40% of species) overlap with oceanic taxa from coastal rainforests, with roughly 70% of IWB taxa restricted to ITR forests, declining further northward and southward in the IWB due to colder and drier conditions, respectively [34]. Within the ITR some arboreal lichens are found predominately in the wettest and oldest (>250 years old) primary forests [35], where they provide food for the Southern Mountain caribou (*Rangifer tarandus caribou*) that shelters in these forests during winter [36]. This caribou is listed by the Committee on the Status of Endangered Wildlife in Canada [37] as threatened in Canada and endangered in BC. Primary forests within the ITR also support many old-growth associated birds (Supplementary File S1), and "sensitive" salmonid species that find refugia within intact watersheds with low road densities [38–40] (Supplementary File S1). In the southern portion of the study area, rainforests escaped glaciation, providing Pleistocene refugia for moisture-seeking vascular plants and bryophytes [11,26,27].

A published dataset on potential lichen habitat [36] provided criteria for identifying the wettest and oldest Interior Cedar Hemlock forests (>250 years old) as high-quality lichen habitats within the BC Vegetation Resources Inventory. Bird species (*n* = 28, Supplementary File S1) that are likely to be associated with old-growth forest were identified by Birds Canada using data and distribution models from the Atlas of Breeding Birds of British Columbia, 2008–2012 (Table S1). Importantly, the bird distribution models combine topographic factors (e.g., slope, aspect, elevation, latitude and longitude) with species detections and survey effort information (number of hours) to predict the probability of observing for a species after 20 h of survey effort within a 10-km cell (an "Atlas Square"—the unit of measure in the British Columbia Atlas—in which participants were asked to complete 20 h of surveying for birds). A Generalized Additive Model (GAM) was used to fit the model across the entire land area of BC. These resulting Probability of Observation ("PObs") raster layers can be thought of as annotated range maps, depicting not only the range of a species but also the likelihood of finding it within a given part of its range after a standardized search period. The models are therefore sensitive to the particular characteristics and requirements of each species (e.g., rare versus abundant; lo-

calized versus widespread; low versus high elevation, etc.), making it possible to combine guilds of species into a single map depicting the overlap of all species. Additionally, the layer for fish species (*n* = 6, Supplementary File S1) sensitive to human disturbances was obtained from a report by the British Columbia Ministry of Environment and Climate Change Strategy with data on their presence in each watershed obtained from the BC Conservation Data Centre (Table S1). The area of potential habitat for woodland caribou has been identified by Environment Canada [41] (Supplementary File S1, Table S1). Core caribou habitat is identified as possessing ecological conditions necessary for populations of Southern Mountain Caribou to carry out life processes, including overwinter survival and reproduction [38].

## 3. Methods

To address what primary forest remains, where it occurs, and how fast it and the species associated with these forests are being depleted, we examined the condition of the IWB and ITR at multiple scales from bioregion (IWB) to subzone (ITR) to watershed. We determined this by comparing potential primary forest to amount and type of anthropogenic disturbances, including the full array of disturbances (e.g., roads, development) and logging (cut blocks). Importantly, IWB and ITR general distribution maps have been published via the Vegetation Resource Inventory for the BC province, but it does not include lands converted to development, agriculture, natural meadows, or water (Table S1). It is a somewhat general zone dataset. We used the term "potential" to describe these lands.

### 3.1. Anthropogenic Disturbances

For the forest impact and analysis assessments in the Red-listed Ecosystem (RLE) Criteria, we overlaid three timber harvesting datasets published by the Ministry of Forests, Lands, Natural Resource Operations and Rural Development: Consolidated Cutblocks [42]; Results Openings [43]; and Forest Tenure Cutblock Polygons FTA 4.0 [44] (Table S1). The union operation was performed using a 50-m XY tolerance to equate near-identical polygons. Due to discrepancies in areas of overlap among the three inputs, layers were prioritized in the order listed above. Logging units were labeled with the decade the operation occurred. Cutblocks were summarized (total and mean area) by decade starting in the 1930s thru 2010s; 137,540 ha contained no origin date and thus were classified as null in the analysis of decadal logging rates.

The anthropogenic fragmentation layer was created by merging: (1) timber harvest cut blocks (from the three cutblock datasets), roads, transmission lines, and railways, (2) agricultural land data by the BC Agricultural Lands Commission, and (3) legally defined municipalities by the BC Ministry of Municipal Affairs and Housing, all available from the BC Ministry of Forests, Lands, Natural Resource Operations and Rural Development (see Table S1 for links to all datasets). To account for edge effects, buffer distances were used around cut blocks (100-m); lanes, trails, alleyways, driveways, unclassified roads (10-m); railways, ramps, local, creation, resource, restricted, runway, service, and strata roads (30-m), and arterials, collector roads, highways, freeways (60-m). Data were rasterized for FRAGSTATS [45] to calculate core percentage of landscape for potential and current forest cover using a 90-m (3 pixels) moving window.

Potential IWB and ITR was estimated by adding current forest cover and those forest areas logged or converted to other land uses. Potential forest cover is shown in Figure 1 and likely to support naturally growing forests under existing environmental conditions. IWB and ITR forest loss was calculated by subtracting potential from current forest cover.

### 3.2. Watershed Analysis (Biotic and Abiotic Factors)

We clipped a total of 3731 watersheds from the standardized Freshwater Atlas of the BC [25] that intersected the boundaries of the IWB (Table S1). We then used the watershed as the spatial unit of analysis to assign spatial extent and degree of impacted area for abiotic and biotic factors for the RLE rankings and for identifying localized areas for protection and

restoration. RLE rankings were informed by accessing existing protected area coverages (combined national, provincial, and private conserved lands) for determining conservation status at the watershed scale [46]. All datasets were processed and uploaded into a gallery on Data Basin (https://databasin.org/galleries/be2d892a33344997963c12a8e521 bd15/ accessed on 10 June 2021) for public access except for sensitive fish locations due to agreement with the BC government.

### 3.3. RLE Criteria

We selected RLE criteria based on applicability to our study area objectives and datasets available for analysis. We used the published thresholds in Bland et al. [12] to define status rankings from Vulnerable (VU) to Critical (CR) (Table 1) with all rankings summarized in a final assessment.

**Table 1.** IUCN Red-listed Ecosystem subcriteria/criterion used in the Interior Wetbelt and Inland Temperate Rainforest of British Columbia, Canada. Adapted from Bland et al. [12].

| Subcriteron/ Criterion | Extent (%) | Severity ≥80% | Severity ≥50% | Severity ≥30% | Critical (CR) | Endangered (EN) | Vulnerable (VU) |
|---|---|---|---|---|---|---|---|
| A1: Reduction in geographic distribution over the past 50 years | | | | | ≥80% | ≥50% | ≥30% |
| C1: The past 50 years, based on change in an abiotic variable affecting a fraction of the extent of the ecosystem and with relative severity, as indicated by the status columns | ≥80 ≥50 ≥30 | CR EN VU | EN VU | VU | | | |
| D1: The past 50 years, based on change in a biotic variable affecting a fraction of the extent of the ecosystem and with relative severity, as indicated in the status columns | ≥80 ≥50 ≥30 | CR EN VU | EN VU | VU | | | |
| E: Quantitative analysis that estimates the probability of ecosystem collapse within 50 years | | | | | ≥50% in 50 years | ≥20% in 50 years | ≥10% in 100 years |

## 4. Results

### 4.1. RLE A1: Reduction in Geographic Distribution (Spatial Analysis)

The RLE A1 assesses status via geographic distribution of an ecosystem based on spatial and temporal indicators. Using the combined impact layer, anthropogenic disturbances totaled 5.6M ha (34%) of the study area; 3.2M ha (57%) of which was from clearcut logging (Figure 2). The largest (39%) reductions in potential forest from all anthropogenic sources occurred within the ITR (Table 2).

Approximately 1.6M ha of IWB and 235,449 ha of ITR are within provincial parks, representing 24% and 29% of remaining IWB and ITR, respectively (Figure 2 above, hatched). Notably, 570,746 ha (71%) of remaining ITR is outside the protected area network where it is vulnerable to additional anthropogenic disturbances (mainly logging).

Loss of core primary forest areas from anthropogenic disturbances was greater at 70% (IWB) and 95% (ITR), increasing the RLE rankings from none to endangered, and vulnerable to critical, respectively (Table 2). Anthropogenic disturbances were mostly concentrated in specific watersheds within the IWB overall (darkest red, Figure 3C).

### 4.2. RLE A1: Reduction in Geographic Distribution (Temporal Analysis)

This part of subcriterion A1 focuses on the temporal losses from logging as the dominant anthropogenic stressor. From the 1970s-on, logging rates increased dramatically, nearly doubling from 5.3% in the 1970s to 10.2% in the 2000s (Table 3). Thus, we chose 50 years for all subsequent assessments of our study area. Notably, the mean area of cut-

blocks in the 1930s–1960s was approximately 2.5× larger than post 1960s cutting; however, the total area logged was much greater from 1960s-on, resulting in cumulative fragmentation even though cut blocks were smaller. Logging was initially concentrated in the lowlands in the most productive ITR areas followed by expansion into specific watersheds over time (not shown at this scale). Areas in proximity to mill centers (e.g., Prince George) were especially impacted.

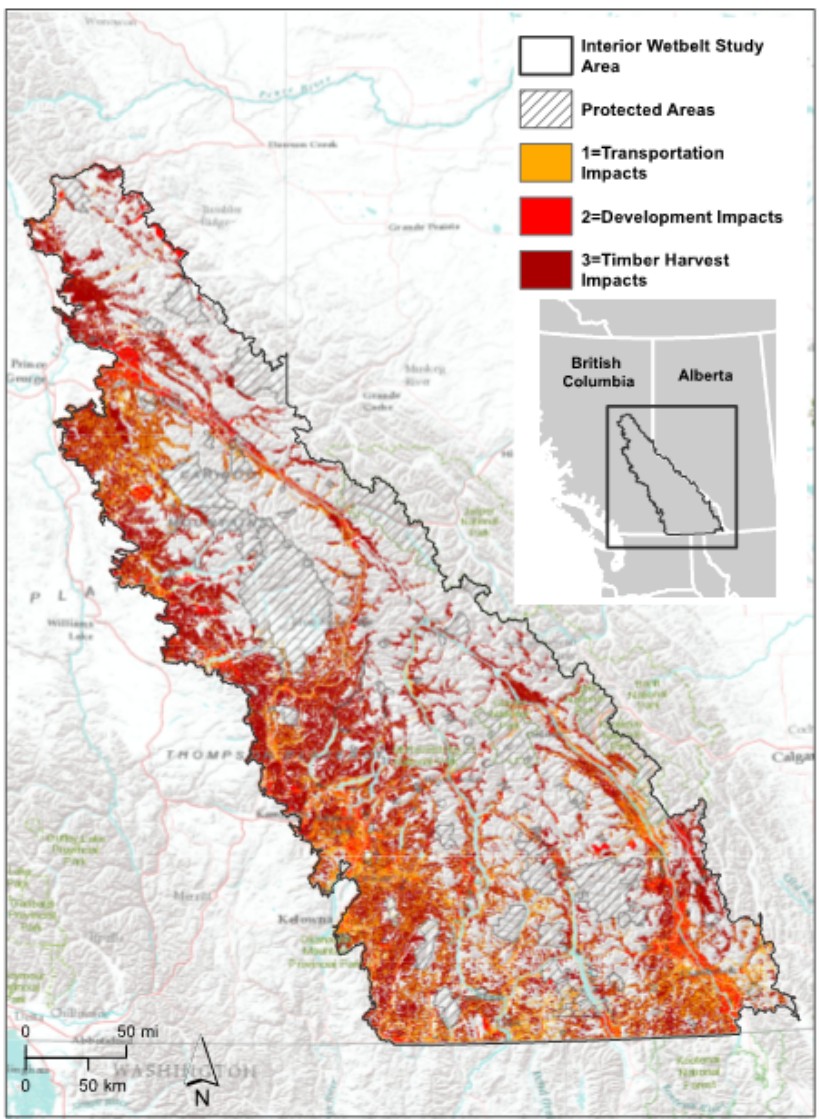

**Figure 2.** Cumulative impacts and parks (hatching) within the Interior Wetbelt Study area, BC, Canada.

**Table 2.** Potential, remaining, and core primary forest in the Interior Wetbelt and Inland Temperate Rainforest, British Columbia, Canada for Red-listed Ecosystem subcriterion A1. Rankings = VU, vulnerable; EN, endangered; CR, critical based on thresholds in Table 1. Adapted from [12].

| Type | Potential Primary Forest (ha) | Remaining Primary Forest (ha) | Reduction (%) | Status | Remaining Primary Core Forest (ha) | Reduction (%) | Status |
|---|---|---|---|---|---|---|---|
| Interior Wetbelt | 9,430,236 | 6,747,906 | 28 | None | 2,828,200 | 70 | EN |
| Inland Temperate Rainforest | 1,330,198 | 806,195 | 39 | VU | 59,810 | 95 | CR |

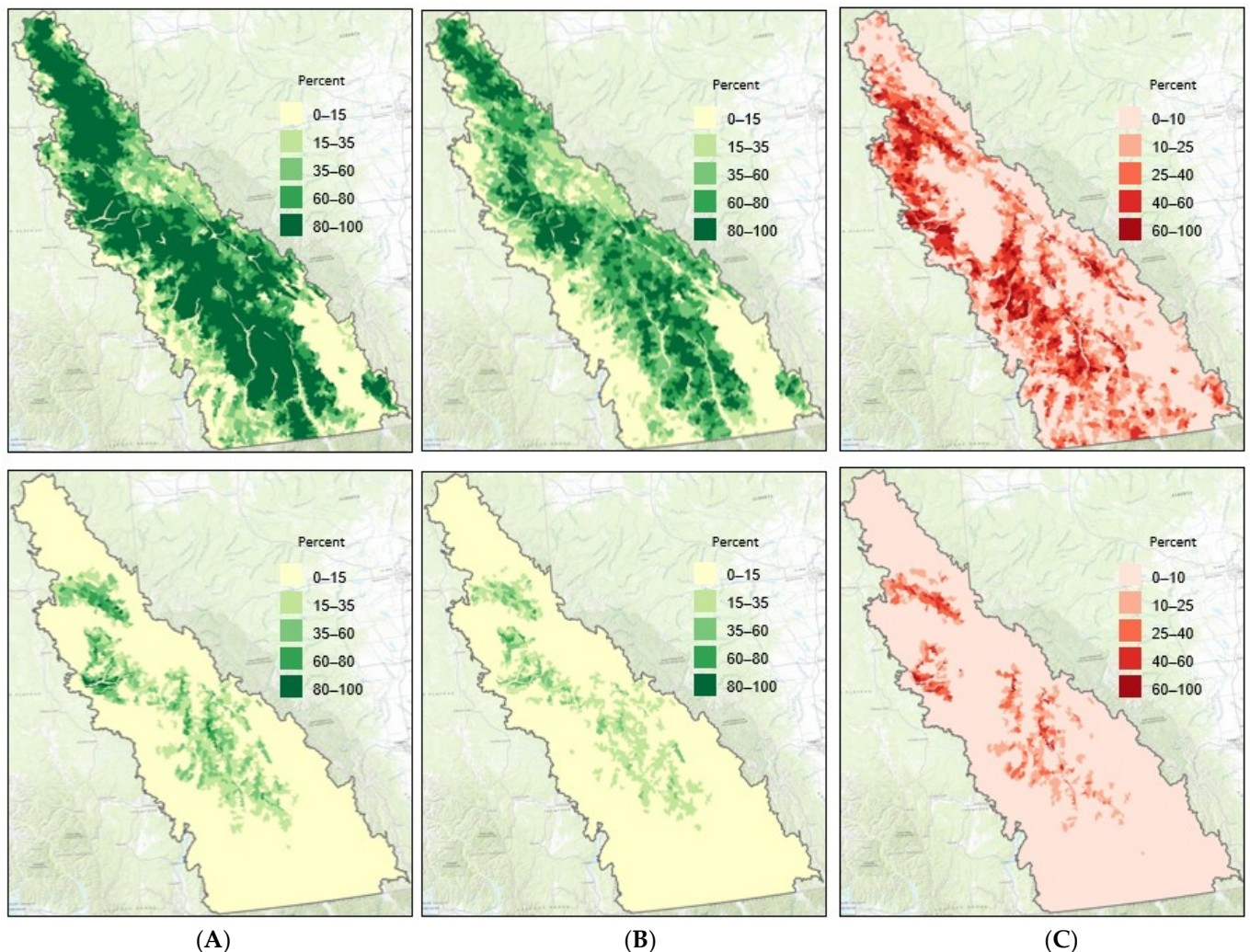

**Figure 3.** Potential primary forest (**A**), current forest (**B**), and forest loss (**C**) in the Interior Wetbelt British Columbia (top panel) and Inland Temperate Rainforest (bottom panel), Canada represent percent cover at the watershed scale. Losses were based on the total anthropogenic disturbance layer.

**Table 3.** Logging by decade and cutblock size for the Interior Wetbelt study area, British Columbia, Canada. Null refers to missing cut block dates that could not be assigned to a particular decade. Percent refers to rate of logging for that decade.

| Decade | Number of Cut Blocks | Mean Cutblock Area (ha) | Logged Area (ha) | Percent of Forest Area Logged |
|---|---|---|---|---|
| unknown (null) | 4921 | | 139,414 | |
| 1930s | 7 | 47.4 | 332 | 0.00 |
| 1940s | 59 | 53.3 | 3142 | 0.03 |
| 1950s | 252 | 58.9 | 14,849 | 0.16 |
| 1960s | 3991 | 49.6 | 198,057 | 2.14 |
| 1970s | 14,065 | 34.2 | 480,727 | 5.30 |
| 1980s | 19,879 | 31.4 | 623,791 | 7.30 |
| 1990s | 27,194 | 18.2 | 494,608 | 6.20 |
| 2000s | 35,560 | 21.4 | 762,416 | 10.20 |
| 2010s | 19,889 | 23.8 | 472,893 | 7.04 |

*4.3. RLE C1: Environmental Degradation Abiotic*

Karst was assessed by focusing on watersheds with ≥10% of this feature to determine spatial extent for the RLE C1 thresholds. There were 1410 watersheds totaling approxi-

mately 6.26 million ha (mean area x total number of watersheds with karst), which was around 38% of the study area. Of these, 1154 (82%) were impacted by anthropogenic disturbances and the average percent human disturbed was 38%, which resulted in an overall REC C1 ranking of vulnerable [12].

### 4.4. RLE D1: Environmental Degradation Biotic

To estimate the extent of the biotic factors impacted, we set the reference level at ~10.2M ha and 1.33M ha for primary IWB and ITR, respectively, for what was available prior to historic logging (>50 years ago) using the Agricultural Land Reserves and Municipalities datalayers (Table S1). We then calculated the percent area affected against this reference using the anthropogenic disturbance layer. Doing so, resulted in three of the four biotic factors near or above the 80% extent impacted threshold (Table 4). Severity was based on the component of the impacted extent completely lost plus the average impact (expressed as percent of watershed that was intact) of the remaining occupied area. For example, 3.9M ha of original caribou core habitat was lost completely, which made up 43% of the original extent impacted. Of the remaining 5.1M ha (57% of the area), the average percent degraded was 28.25. Thus, for caribou a severity index of 59% was calculated [43% + (28.25% × 57%) = 59%], putting it in the EN category, which aligns with the overall endangered status of the Southern Mountain Caribou population.

**Table 4.** Biotic factors and rankings for Red-listed ecosystem Criterion D1 or watersheds within the Interior Wetbelt, British Columbia, Canada. Adapted from [12].

| Biotic Factor & Original Extent (ha) | Extent Remaining Intact (ha) | Extent Impacted (ha) (%) | Severity Index [†] (%) | Ranking |
|---|---|---|---|---|
| Woodland Caribou Core Habitat 10.2 M ha | 1,177,470 | 9,022,530 (88) | 59 | EN |
| Old-Growth Bird Species 10.2 M ha | 2,213,820 | 7,986,180 (78) | 59 | VU |
| Sensitive Fish 10.2 M ha | 546,840 | 9,653,160 (95) | 60 | EN |
| Old-Growth Lichens (ITR only) 1.33 M ha | 154,350 | 1,145,650 (88) | 81 | CR |

[†] Severity Index = (proportion of the landscape where the factor was eliminated × 100) + (proportion of the landscape still containing the factor × average percent of the remaining watersheds degraded).

In addition to the RLE rankings, our maps of key biotic factors can be used to assign conservation and restoration priorities at the watershed scale for local conservation planning. For instance, watersheds with high percent cover for specific biotic factors (e.g., dark blue core caribou habitat, dark blue old growth bird richness) could act as reserves, while lower values (lighter colors having less percent cover) could be assigned restoration priorities for those biotic factors (Figure 4).

### 4.5. RLE E: Probability of Ecosystem Collapse

We used lower and upper bound decadal logging rates from the 1970s–2010s (see Table 3) to assess the probability of ecosystem collapse of remaining IWB and ITR core primary forest in the next 50 years. We assumed core forests were the best indicator of ecosystem integrity and that if only small patches remain, their ecosystem functions are so fragmented that functional collapse of primary areas is imminent (e.g., system-wide edge effects). We assumed what remains (%) of primary core areas corresponds to the probability of ecosystem persistence over time to align RLE E to the conditions of our study area.

With 2,828,200 ha of core IWB primary forest remaining, 22% to 57% may gone within 50 years at the lower (5.3%) and upper (10.2%) decadal logging rates, respectively. This rate corresponded to an EN to CR ranking, depending on logging estimates. With only 59,810 ha of core ITR primary forest remaining, intact ecosystem functions may collapse (zero left) within 18 years (lower bound) to 9 years (upper bound) yielding a CR status ranking [12].

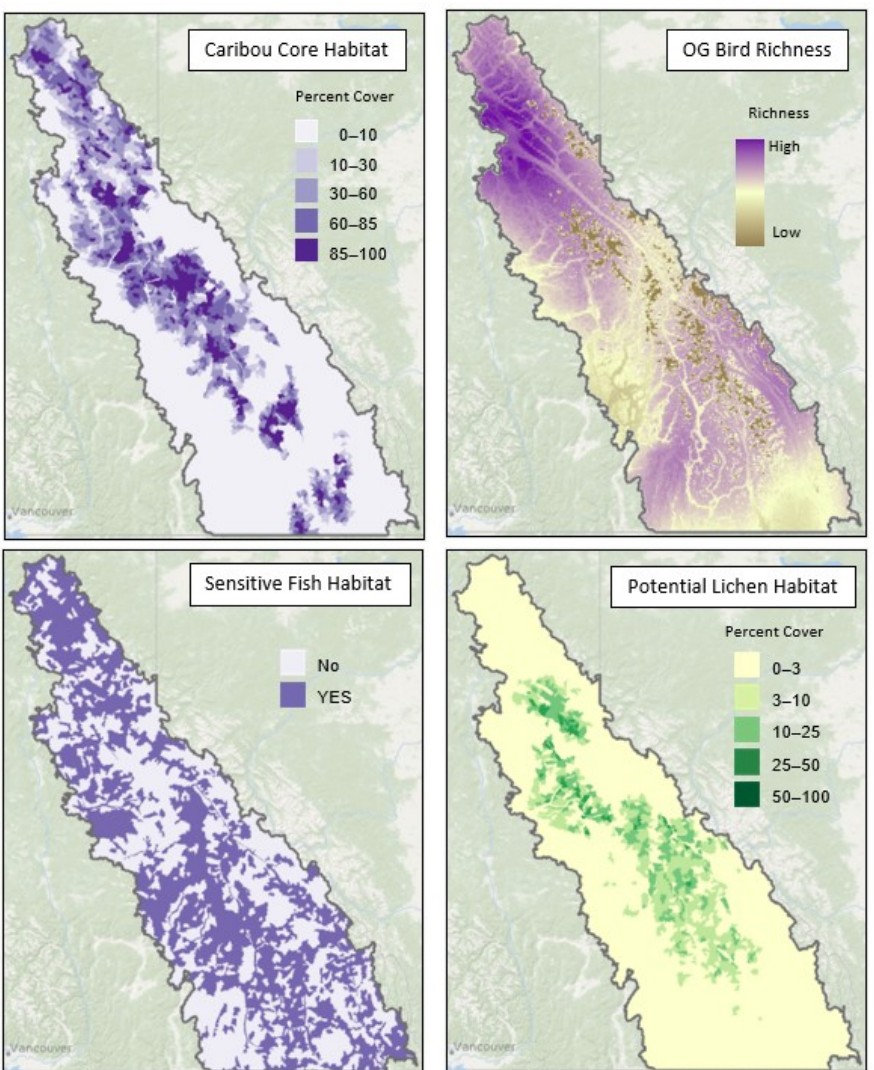

**Figure 4.** Caribou core habitat, old-growth bird richness concentrations, sensitive fish locations, and potential lichen habitat within the Interior Wetbelt, British Columbia, Canada.

*4.6. Uncertainties*

Uncertainties in RLE E are due to logging rates over a 50-year period that otherwise fluctuate annually based on timber demand, access and costs to remaining sites, emerging technologies (e.g., woody biomass pellet manufacturing is increasing in the IWB), natural disturbances resulting in salvage logging, other anthropogenic disturbances that were not included, and potential changes to forestry regulations and policies that could either increase or decrease overtime. Most notably, future logging rates may not be linear and instead may follow a sigmoidal distribution until most if not all accessible old growth is gone [15].

## 5. Discussion

We applied the IUCN Red-listed ecosystem criteria [12] to assess the overall condition of primary forests within the IWB and ITR using a multi-scaled approach. Status rankings ranged from None to EN for the IWB bioregion but were elevated for core primary forests to EN (Table 5). For the ITR, rankings ranged from VU to Collapse with elevated status to CR for core primary forests. For the abiotic factor (C1: karst), VU was assigned, while VU was assigned to Old-Growth Birds, EN to Southern Woodland Caribou and Sensitive Fish, and CR to Old-Growth Lichens. Using the highest-level criteria rankings [12,13] results in

an overall status assessment of EN for the IWB (critical if higher bound logging levels are used) and CR-Collapse for the ITR.

**Table 5.** Combined Red-listed Ecosystem Subcriteria/Criterion A1–C1 and E for the Interior Wetbelt and Inland Temperate Rainforest, BC. Table 1 is split from the Table because the columns were different. NA—not applicable; EN—Endangered, VU—Vulnerable, CR—critical. Adapted from [12].

| Subcriterion/Criteria | Interior Wetbelt | Core Interior Wetbelt | Inland Temperate Rainforest | Core Inland Temperate Rainforest |
|---|---|---|---|---|
| A1 | None | EN | VU | CR |
| C1 | NA | NA | VU | NA |
| E | EN-CR [†] | NA | Collapse | NA |
| **D1: Biotic Factor** | | | **Ranking** | |
| Caribou | | | EN | |
| Old-Growth Birds | | | VU | |
| Sensitive Fish | | | EN | |
| Old-Growth Lichens | | | CR | |

[†] Ranking depends on lower or upper bound of logging rates projecting out 50 years.

Most importantly, the collapse of ecosystem functions of core (unfragmented) ITR primary forest could occur within just 9 to 18 years depending on future logging rates. Notably, the ITR has a natural dendritic distribution in river corridors and toe slopes that may not have historically been distributed in large contiguous blocks [28]. Nonetheless, the type and degree of anthropogenic fragmentation (e.g., clearcuts, roads, transportation corridors, agriculture) differs markedly from natural landscape heterogeneity and thus stepped-up conservation is vital to forestall imminent collapse of core primary functions (e.g., nest predation along edges; increased wind, tree blow down, dessication, and microclimatic changes in general have been reported along forest edges that could prove detrimental to climate—and fragmentation-sensitive species like old-growth birds and lichens).

Several taxa evaluated in Subcriterion D1 are associated with primary forests threatened at the province level, some of which are the main food source (old-growth lichens) of the endangered caribou that likewise depend on primary forests in the winter for food and shelter [36,37]. Old-growth associated bird species like the northern goshawk (*Accipiter gentilis langili*) (included in the bird richness layer, Supplementary File S1) are threatened in BC due mainly to logging [47]. Sensitive fish like bull trout (*Salvelinus confluentus*) and Chinook salmon (*Oncorhynchus tshawytscha*) (included in our sensitive fish layer, Supplementary File S1), both U.S. federally threatened species, are known to decline as sedimentation and stream temperatures increase with logging, forest fragmentation, and high road density [48,49].

We documented that logging rates—the dominant anthropogenic disturbance—nearly doubled during the 1970s–2000s from 5.3% to 10.2% per decade, declining a bit in the 2010s (7.04%). More recently, however; old-growth logging approvals by the BC government have increased by over 40% from 2020 to 2021 [50], presumably reflecting strong demand for timber and a shift away from interior pine towards IWB spruce especially in upslope areas. Should this increase continue, it likely means the upper bound logging levels are reasonable for the foreseeable future at least for the higher elevation snow forests where spruce is mainly distributed that are prone to periodic insect outbreaks and subsequent post-disturbance logging.

Notably, prior to 1960s there may have been more selective logging than current clearcut logging practices, but those activities were largely undocumented spatially and would not show up in the data layers. Minimum retention within cutblocks under current guidelines is just 3.5% [51]. In practice, retention areas are often at the edge of a cutblock, (some up to 400 ha), made up of mostly shrubs, and can be single-trees scattered throughout the unit. We also elected not to include Old-Growth Management Areas (OGMAs) identified by the BC government during unit planning because of their precarious status in policy. The boundaries of OGMAs are routinely modified to permit logging and often

do not contain primary forest. Moreover, very few OGMAs contain forest with interior condition [52].

Our findings on primary forest loss corroborate concerns raised by Price et al. [15] who found that while 26% of forest area (13.2M ha) in the province is in old growth condition, only ~3% of the old-growth (or <1% total forested area) supported the largest trees on the most productive sites (high site index). The largest trees in our study area are within the ITR where logging has mainly concentrated. Matsuzaki et al. [17] recommended conserving carbon-rich primary forests wherever possible within the ITR and high-retention logging to maintain carbon stocks in all other forests. Craighead and Cross cited in [11] recommended protecting 45% of the ITR to achieve representation and viability of large carnivores and caribou. The Canadian government has pledged to protect 25% to 30% of lands and waters by 2025 and 2030, respectively [16,53].

While 24% and 29% of remaining IWB and ITR, respectively, in our study area is included in regional protected areas, most of the core functionality of the IWB (70%) and ITR (95%) is gone and 71% of remaining ITR is outside protected areas. Our findings indicate even the modest Canada Target 1 may not be enough to prevent imminent core collapse of the ITR within a decade or so. Consequently, full protection of remaining primary forest is especially warranted. Restoration should also be targeted at the watershed scale by allowing logged areas time to recover older forest characteristics, and by regenerating the cleared areas of roads to restore connectivity across watersheds and core ecosystem functions. Such measures may be important in climate adaptation strategies [29] particularly since several species in our study area had Moderate (M) to High (H) vulnerability rankings based on climate sensitivity and presumed adaptive capacity, including caribou (M–H), northern goshawk (H), Coho (H), Chinook (H), sockeye (H), bull trout (H), and sturgeon (M–H) [54].

Despite using different methodologies and datasets, our RLE rankings for the IWB and ITR fell within the ecoregion rankings of WWF [21] in the areas of boundary overlap: Central BC Mountain Forests (Vulnerable) and Okanagan Dry Forests (Critical), indicating the robustness of the RLE criteria. However, we note uncertainties in using the RLE criteria. For instance, spatial distribution thresholds in Criterion A and B differed only in whether the indicator in our study area was based on minimum polygon size, so we chose not to analyze Criteria B given we already provided distribution status. The status of abiotic factors (Criterion C) was difficult to assess for most features given geological features like mountain ranges remain present even under development pressures, which is why we chose karst because of its association with rare plants. Criterion D (biotic) included multiple taxa and use of species richness indices that can obscure individual species trends, which warrant follow up study to tease out which species within the groupings are responding to loss of primary forest. Assessing Criteria D also proved challenging given assumptions regarding extent and severity of impacts to fit the data. Additionally, we used remaining primary forest core, not potential, to assess status as calculating effects of natural disturbances on primary core areas prior to logging was out of our project scope. However, at least for the ITR large-scale natural disturbances are on millennial time scales [31] and thus it is reasonable to assume most of the forest was primary prior to industrial logging. Historic logging estimates for our study area may also be conservative given the large number of cutblocks that had no date assigned and could not be used in the logging decadal rate estimators.

## 6. Conclusions

Collapse occurs when ecosystems are pushed to the brink from cumulative anthropogenic disturbances over a rapid time period. Our status rankings in summation indicate that at least for the ITR, it is on a trajectory toward what may very well be the most imperiled temperate rainforest on Earth. Unfortunately, the BC and Canada governments have continued to neglect the region in conservation strategies compared to the Great Bear Rainforest as abiotic and biotic factors in the IWB have lost ground with the rate of logging approximately doubling (1970s–2000s). Follow up World Heritage nomination

of remaining primary forests in collaboration with First Nations may help link the RLE rankings to international conservation pressure given the Canadian and BC government's reluctance to act. Additionally, a Red-listed assessment of lichens associated with primary old-growth forest may help with the identification of the area as a Key Biodiversity Area.

We believe our findings about the threat status of the IWB are relevant to the US portion as well [11] with the notable exception of lichens that are less diverse in the southern extent of the bioregion where it is drier [18,19]. Given the precarious status of the IWB, there is an urgent need to greatly improve forest management and elevate the conservation status of remaining primary forests in both the US and Canada, particularly core areas within the ITR. The critical need for rapid action is revealed by the core functions of the ITR are within a decade or two of collapse.

**Supplementary Materials:** The following are available online at https://www.mdpi.com/article/10.3390/land10080775/s1; Supplementary File S1: Data analysis methods used in the Interior Wetbelt, BC study design; Table S1: Datalayers summarized to the watershed reporting unit for the BC Interior Wetbelt study area. All datasets posted in a gallery portal on (https://databasin.org/galleries/9ac9d9d1b0954c9aa2bfa667081e3723/) and all links active accessed on 29 May 2021.

**Author Contributions:** All authors were involved in the writing and approval of this manuscript. D.A.D. (project lead) was involved in all aspects of the research including conceptualization, methodology, field work, formal analysis, investigation, data curation, and writing; J.R.S and R.D. performed formal data analyses including managing the databasin.org project portal; M.C. and J.R.W. provided field support and local investigative knowledge; B.M. provided project conceptualization as part of a global set of primary forest case studies; H.K. provided manuscript and formal analysis support as part of a global set of primary forest case studies; D.C. and A.C. provided formal expertise and analysis support on lichens and birds, respectively. All authors have read and agreed to the published version of the manuscript.

**Funding:** This research was supported by a grant to BM from an anonymous charitable organization that neither seeks nor permits publicity for its efforts and was not involved in any aspects of this project.

**Institutional Review Board Statement:** Not applicable.

**Informed Consent Statement:** Not applicable.

**Data Availability Statement:** All data in this study were posted for public access except where noted https://databasin.org/galleries/9ac9d9d1b0954c9aa2bfa667081e3723/ accessed on 10 June 2021.

**Acknowledgments:** We are thankful to Jessica Leonard for technical support during this project in the preparation of datasets and figures for posting on https://databasin.org/galleries/be2d892a33344997963c12a8e521bd15/ accessed on 10 June 2021.

**Conflicts of Interest:** The authors declare no conflict of interest.

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
