# Peer review of "Red-Listed Ecosystem Status of Interior Wetbelt and Inland Temperate Rainforest of British Columbia, Canada"

_land, doi:10.3390/land10080775_

Round 1
Reviewer 1 Report
Review of DellaSala and others. “Red-listed ecosystem status of interior wetbelt and inland temperate rainforest of BC, Canada”.
June 30, 2021
This paper assesses the conservation status of BC’s inland wetbelt and temperate rainforest ecosystems using selected IUCN red-listed ecosystem criteria. These ecosystems are globally rare, are exploited by industrial-scale timber harvest, and are much less familiar to most people than BC’s coastal temperate rainforest; hence the assessment is highly relevant. BC’s provincial government is currently designing a new approach to managing old growth forest; hence the assessment is also timely.
Overall, I believe the paper important and interesting. The writing could be tightened throughout—and a close reading is needed to catch typos and inconsistencies. I have some questions and suggestions—some substantive, some trivial—for the authors. I present the conceptual and substantive topics first and then list minor suggested edits.
- Clarify terminology. This paper focuses on “primary” forest. In the introduction, I’d like to see a description of primary forest and a sentence or two that describes the relationship between old growth and primary and natural young. What is “potential” forest? In Figure 1, forests are mapped as “potential”. Given no definition up to this point, I was confused as I thought the extent of IWB and ITR was already mapped and published. Does “potential” simply mean IWB/ITR of all ages, managed or primary? Later (Table 2), “potential” seems to imply forest of all ages and origins. It’s a confusing term: that’s not “potential…forest”; it is forest by most definitions. Figure 3 uses “potential primary forest”, “current forest” and “forest loss”. What is forest “loss”? I don’t dispute that “managed forest” or “plantation forest” differs from “primary forest”, but managed forest differs from land conversion to urban area or ecological regime shift to grassland, both indisputably “loss”. From an ecological perspective, some earlier logging with high retention may be developing structure that resembles primary forest structure. I recommend defining terms in the introduction, using terms consistently throughout and—most importantly—considering changing the language. Maybe focus on “primary” vs. “non-primary” or “remaining primary” vs. “managed forest” or something similar. If you believe that plantation forests can never recover to their primary status, then they’ve lost their “potential”.
- Add nuance to description of clearcuts. There’s use of “clearcut logging” throughout. The datalayers, however, do not provide information on retention within clearcuts. The older cutblocks will likely have retained a fair bit of structure—regulations to take all trees didn’t come into effect until mid-century, and most early harvest was high-grading. Over time, some of these stands may have recovered to have good structure that approaches primary forest (my experience is with coastal temperate rainforest, so I may be wrong for interior rainforest, but on the coast, I have seen high-graded stands with good structure and populations of epiphytic lichens and birds that resemble naturally-disturbed primary stands more than clearcut stands). More recently, harvesting leaves retention (I agree that the amount is relatively low) that should be acknowledged (some datalayers have retention removed; others don’t). I think it’s important to discuss retention in methods and discussion.
- Change buffers as cutblocks age. The methods suggest that buffers differ among edge types (line 216 and on state that cutblocks have a 100-m buffer while linear features have lower buffers varying from 10 – 60m), although the abstract lists all buffers as 100m. Edge effect depends on the variable of interest. What edge effects and criteria did you use to determine buffer width? Please add a description of the effects you considered and a rationale for the different widths. We generally use 100m on wide high-use roads (including mainlines) as they are hard edges that remain over time. I’m interested in why you would include 100-m buffers around cutblocks of all ages. As vegetation grows, the edge effects will diminish. What types of effects do you envision will impact primary forest from a mature managed forest? The assumption of a continuing 100-m edge effect over time may inflate the impact on core ecosystems. I would like to see the data re-analysed to include a reduced edge effect over time (perhaps 100-m up to age 30, then no buffer; or 100-m to 20 and 50-m from 20 – 40 years; or something similar, with a rationale). It will at least be important to include such an analysis in the paper as a sensitivity analysis. Given that the loss of core area is a critical part of the story, it is crucial that this analysis is fully defensible. I believe that using 100-m buffers for mature adjacent forest is not defensible.
- Clarify BEC subzones included. Line 116 mentions the wc, vc and m subzones of the “Sub-Boreal Engelmann Spruce-Subalpine fir…”. I’m not sure what you mean here (this could be trivial and a simply typo). Sub-Boreal Spruce is a different zone than the ESSF. As far as I’m aware, SBS isn’t part of the IW or ITR (at least it’s not listed in the 2019 Coxson, Goward, Werner biome description). Certainly, the moist SBS subzones aren’t included. Also, you might want to ensure that you’re referring to the zones correctly—check the references you cite there (e.g., Engelmann Spruce – Subalpine Fir).
- Remove RLE B1. In this version of the paper, A1 and B1 measure the same thing (as you note in the discussion). B1 as described in the methods is a smallest convex polygon of locations. Your analysis does not do this. Either follow the methodology as stated, and give the size of the polygon, or remove this criterion. There is no benefit I can see in presenting the same data two ways, and a potential cost in that it appears to violate independence.
- Clarify RLE C1. The methods suggest that ranking depends on extent and severity. The results only mention the number of watersheds with any anthropogenic disturbance. Did you assess severity? If a watershed had a small 5-ha clearcut over the ridge top, does this count as an “impacted” watershed? More transparency is needed here.
- Clarify severity for RLE D1. In calculating the average percent degraded of the not-completely-degraded watersheds, did you consider non-normality? At least it would be good to know as averages can be quite non-representative in some cases (in which case, you could use median perhaps).
- Add simple description of methods for birds. Appendix S1 is too technical for a general audience. I’ve analysed bird data, and I have trouble understanding it (e.g., how is a PObs raster a species?). There’s no reason that the body of the paper can’t include a paragraph that describes methods simply. I question using species richness—some species listed are much more sensitive than others; for example, BCCH are common garden species, RBSA are associated with deciduous in some areas, while BRCR are well-established old growth specialists and NOGO need 60% foraging area to be at least mature. Some species listed do reasonably well with levels of retention even as low as those in policy. It’s difficult for me to review this section (even though I’ve published bird studies) because I don’t understand it. I’m guessing if I have trouble, others will have more difficulty.
- Increase discussion of RLE D1 relative to natural expected habitat. Caribou, old growth birds and specialised epiphytic lichens need old primary forest, not just primary forest. Thus, the comparison should ideally consider age as well as origin. It would be possible to use natural disturbance estimates to estimate the amount of old forest expected and use this in a sensitivity analysis. The discussion simply states that comparing to natural amounts was out of scope—I think a more fulsome discussion, including the ramifications of your assumptions (for example, that the proportion of old is the same within the remaining area as within the total managed plus primary area) is needed here.
- Discuss assumption of persistence = remaining primary core forest. This is a big assumption and needs discussion. I understand that you’re looking for a way to mesh the data with the IUCN thresholds, but you should at least discuss other risk measures (e.g., sigmoidal 30-70% curves). Are there no ITR areas large enough in protected areas? I would say that uncertainties aren’t just due to unpredictable logging rates but also to the linear assumption.
- Compare various approaches to determining ecosystem status (in the discussion likely). The introduction (para 1, line 42-47) mentions other assessment methods and concludes that they are “not comprehensive enough” for the study area. It doesn’t, however, present evidence of this lack (either here or in the discussion, unless I missed it), beyond noting they’re not identified as hotspots or key biodiversity areas yet. Given the claim (line 85) that this is the first application of RLE criteria for forests in North America, I would like some discussion of a) why other approaches wouldn’t work and b) how well you felt RLE applied. What are the costs and benefits? Would multiple pathways of evidence be useful? Or is this sufficient?
- Add climate change. The discussion would benefit from a climate change paragraph. Many of these species have had vulnerability assessed in a BC Fish and Wildlife Vulnerability database. https://www2.gov.bc.ca/assets/gov/environment/natural-resource-stewardship/nrs-climate-change/adaptation/climate20change20vulnerability20of20bcs20fish20and20wildlife20final20june6.pdf (I could only find the report, not the database, online)
Minor edits
Why are endnotes roman numerals? They’re really hard to read.
Section 1.
Line 62. I question whether primary forests are a province-wide priority in BC. The 2 cited references are both unpublished (there is a published version of reference #13 that should replace that one). While BC has recognised the importance of old growth, they have not recognised the importance of naturally-disturbed primary forests and still use age criteria to define old growth. Provincial government legislation, regulations and policy do not discuss primary forest. This paragraph seems to mix a couple of concepts (BC and Canada’s conservation priorities and the importance of IWB/ITR forest). I’d make it into two paragraphs. It’ll be important to have the comparison of primary and old growth forest (as described in the “clarify terminology” substantive suggestion above) before (or within) this section.
Section 2.3
Line 133. Western redcedars are not “antique forests”; they are trees. Add a sentence on antique forests to the one on ancient trees.
Section 2.4
Line 156-159. Perhaps listing the months for the quarters with stats would help. (e.g., add JJA after warmest quarter if those are the months included).
Line 164. Awkward sentence. Fix.
Section 2.5
Line 177. Typo. “respectfully” should be “respectively”.
Line 186. “moisture-seeking plants, mosses, and hepatics (bryophytes)” is puzzling. All are plants. Mosses are also bryophytes. Clarify.
Line 191. Typo: “form” should be “from”.
Section 3.1
Line 202. Write out RLE in full the first time in the methods. I’d forgotten what it was by that point.
Line 202 and on. These methods are ok, but you could use some friendlier language at least in the first sentence. “the product of a union among…” is unclear. Is “product” a mathematical operation (I think not). Union is correct, but maybe say “we overlaid three forest harvesting datasets…” in the first sentence and keep “union” for the explaining sentence afterwards. Also maybe explain that the three datasets are needed for the most accurate estimate of cutblocks.
Line 209 “thru”??? Canadian spellings for Canadian ecosystems please!
Section 3.2
Line 231. Did you include Old Growth Management Areas in the conservation status? In this area, there are non-legal and legal OGMAs. Although they offer limited protection, they’re worth mentioning particularly as they’re BC’s way of managing old forest.
Line 234. Cool that you uploaded data (I haven’t had a chance to check it).
Table 1. The formatting on this table sucks on the version I have. It’s really challenging to understand. I’d also write out the rankings in full for ease of reading. The “extent and relative severity” bits seem strange—they don’t seem to differ among columns. A footnote would be better.
Table 1. Criterion B1. Is the polygon within the study area or globally? Obviously, boundaries will dictate size. In this case, the ecosystem extends south of the Canadian border.
Section 4.1
Line 247. “,” not “;”
Table 2. Lose the decimals from the %. Just makes it harder to read. I wonder if there’s a way to make the tables clearer?
Table 2. Include the thresholds in the table title or in the table for ease (or at least refer folks to Table 1).
Line 261-3. Update Table # references. This paragraph is important. I’d like the writing tightened and maybe the ranks written out. Even a minor change would help “Loss of core forest areas from anthropogenic disturbances was greater at 70% (IWB) and 95% (ITR), increasing the RLE rankings from none to endangered, and vulnerable to critical, respectively.”
Section 4.2
Line 272. Is cutblock size relevant to the criterion? The criterion seems to focus on total amount (which is consistent with patterns of fragmentation). I’d suggest that instead of noting that cutblocks were 2.5x larger 1930s-60s you focus on the % of forest area logged as xx times more 1970s-2010s.
Table 3. “missing data” seems preferable to “lack of information”. I think filling in the missing cells would be useful to see how much data are missing.
Section 4.3
Table 4 mislabelled as Table 3.
See RLE B1 substantive issue above. Suggest removing this section.
Section 4.4
Line 293. It’s curious that the 1,419 watersheds averaged 4,443 ha—the identical mean size as the 3,722 watersheds mentioned in Table 4 (labelled 3) in section 4.3. Was this the mean size of the karst watersheds or of the total watersheds? If you know the number of karst watersheds, you can calculate the actual area and average size.
Line 296. “REC” should be “RLE” I assume. Please read for typos.
Section 4.5
Line 299 and Table “4”. Reference level is given at 10.2 million ha for IWB, yet Table 2 and section 2.2 say 9.4 million ha. Am I missing something here? If there’s a reason for the discrepancy, please explain.
Table “4” is actually “5”. Check table #s throughout.
Figure 4. Caribou “core” habitat. Best use a different term as this could be confused with “core” (i.e., non-edge) forest.
Figure 4. Consider grey scale for printing.
Section 5.
Improve formatting of Table 5 (should be 6).
Line 374. Use Canadian designations rather than US.
Line 381. True that salvage of pine is down, but demand is spruce everywhere, not just IWB. And definitely upslope in ESSF interior.
Line 385. Ref 13 is now published.

Author Response
Reply to Reviewer 1 – we thank the reviewer for the many helpful comments and suggestions which we have incorporated mostly. Our response to each is indicated below.
Clarify terminology. This paper focuses on “primary” forest. In the introduction, I’d like to see a description of primary forest and a sentence or two that describes the relationship between old growth and primary and natural young.
Reply – we added a definition and citations on lines 45-49, and general reference to old growth (line 78), thank you.
What is “potential” forest? In Figure 1, forests are mapped as “potential”. Given no definition up to this point, I was confused as I thought the extent of IWB and ITR was already mapped and published. Does “potential” simply mean IWB/ITR of all ages, managed or primary? Later (Table 2), “potential” seems to imply forest of all ages and origins. It’s a confusing term: that’s not “potential…forest”; it is forest by most definitions. Figure 3 uses “potential primary forest”, “current forest” and “forest loss”. What is forest “loss”? I don’t dispute that “managed forest” or “plantation forest” differs from “primary forest”, but managed forest differs from land conversion to urban area or ecological regime shift to grassland, both indisputably “loss”. From an ecological perspective, some earlier logging with high retention may be developing structure that resembles primary forest structure. I recommend defining terms in the introduction, using terms consistently throughout and—most importantly—considering changing the language. Maybe focus on “primary” vs. “non-primary” or “remaining primary” vs. “managed forest” or something similar. If you believe that plantation forests can never recover to their primary status, then they’ve lost their “potential”.
Reply – we added lines 247-251 to improve the meaning of potential forest. We agree that different types of disturbance have different degrees of impact – some lands are converted to urban development and agriculture, which are more impactful and for practical purposes permanent. Clearcut logging as applied today in this landscape is less destructive than converting these lands to pavement, but they will be managed as plantations and will not be allowed to grow to full maturity. In that sense, they are lost.
Add nuance to description of clearcuts. There’s use of “clearcut logging” throughout. The datalayers, however, do not provide information on retention within clearcuts. The older cutblocks will likely have retained a fair bit of structure—regulations to take all trees didn’t come into effect until mid-century, and most early harvest was high-grading. Over time, some of these stands may have recovered to have good structure that approaches primary forest (my experience is with coastal temperate rainforest, so I may be wrong for interior rainforest, but on the coast, I have seen high-graded stands with good structure and populations of epiphytic lichens and birds that resemble naturally-disturbed primary stands more than clearcut stands). More recently, harvesting leaves retention (I agree that the amount is relatively low) that should be acknowledged (some datalayers have retention removed; others don’t). I think it’s important to discuss retention in methods and discussion.
Change buffers as cutblocks age. The methods suggest that buffers differ among edge types (line 216 and on state that cutblocks have a 100-m buffer while linear features have lower buffers varying from 10 – 60m), although the abstract lists all buffers as 100m. Edge effect depends on the variable of interest. What edge effects and criteria did you use to determine buffer width? Please add a description of the effects you considered and a rationale for the different widths. We generally use 100m on wide high-use roads (including mainlines) as they are hard edges that remain over time. I’m interested in why you would include 100-m buffers around cutblocks of all ages. As vegetation grows, the edge effects will diminish. What types of effects do you envision will impact primary forest from a mature managed forest? The assumption of a continuing 100-m edge effect over time may inflate the impact on core ecosystems. I would like to see the data re-analysed to include a reduced edge effect over time (perhaps 100-m up to age 30, then no buffer; or 100-m to 20 and 50-m from 20 – 40 years; or something similar, with a rationale). It will at least be important to include such an analysis in the paper as a sensitivity analysis. Given that the loss of core area is a critical part of the story, it is crucial that this analysis is fully defensible. I believe that using 100-m buffers for mature adjacent forest is not defensible.
Reply – the reviewer has a point; however, we prefer not to change buffer widths post analysis to prove something that seems rather obvious to us. Changing the buffer width would indeed affect the amount of core primary forest, if, say we reduced the width to 50-m, then core primary forest would go up. Conversely, if we increased buffer widths to 1-km like some studies (e.g., Ibisch et al. 2016 in our paper, Briant et al. 2010; https://www.sciencedirect.com/science/article/abs/pii/S0006320710003307) then it would disappear all-together for at least the ITR. The classic paper is Chen et al. (1992: https://www.jstor.org/stable/1941873) on depth-of-edge influence for abiotic factors documented up to 137-m in old-growth Douglas-fir forest juxtaposed with clearcuts. Bezzola and Coxson (2020; https://cdnsciencepub.com/doi/abs/10.1139/cjfr-2019-0381?journalCode=cjfr) used a 120-m edge effect for fragmentation analyses that overlapped with our study area. DellaSala’s doctor dissertation documented edge impacts on avian communities >100-m into mature northern hardwood forests of northern Michigan juxtaposed with recent clearcuts. Importantly, around 89% of the mapped clearcut patches in our study area are <50 years old. Mature forest characteristics for which many of the region’s old-growth species depend (e.g., woodland caribou, lichens, and OG birds) take at least 80-100 years with classic old-growth features not present until 250 years (Price et al. 2021). It is therefore likely that an old-growth forest with trees up to 1,600 years juxtaposed next to clearcuts <50 years old is experiencing edge effects at least out to 100-m. If the reviewer insists, we can add this to the uncertainties discussion but we believe this topic is well covered in the literature and in discussion forums like ResearchGate.
Clarify BEC subzones included. Line 116 mentions the wc, vc and m subzones of the “Sub-Boreal Engelmann Spruce-Subalpine fir…”. I’m not sure what you mean here (this could be trivial and a simply typo). Sub-Boreal Spruce is a different zone than the ESSF. As far as I’m aware, SBS isn’t part of the IW or ITR (at least it’s not listed in the 2019 Coxson, Goward, Werner biome description). Certainly, the moist SBS subzones aren’t included. Also, you might want to ensure that you’re referring to the zones correctly—check the references you cite there (e.g., Engelmann Spruce – Subalpine Fir).
Reply – good catch on the SBS layer incorrectly stated as part of the ITR, which was an error on our part in the write up, but has now been removed from the ITR discussion.
Remove RLE B1. In this version of the paper, A1 and B1 measure the same thing (as you note in the discussion). B1 as described in the methods is a smallest convex polygon of locations. Your analysis does not do this. Either follow the methodology as stated, and give the size of the polygon, or remove this criterion. There is no benefit I can see in presenting the same data two ways, and a potential cost in that it appears to violate independence.
Reply – we agree and deleted B1.
Clarify RLE C1. The methods suggest that ranking depends on extent and severity. The results only mention the number of watersheds with any anthropogenic disturbance. Did you assess severity? If a watershed had a small 5-ha clearcut over the ridge top, does this count as an “impacted” watershed? More transparency is needed here.
Reply - As noted in Section 4.4 of our paper and in the footnote to Table 4, we calculated change in extent and severity according to the rubric matrix. For example, the reference habitat area for caribou was 10.2M ha. The amount still intact was 1,177,470 ha. Therefore, the amount of habitat impacted was 88%, which gives us the first extent number in the rubric matrix table. That means the amount of impacted area for caribou was 9,022,530 ha. Of this amount, 3,893,700 ha (43%) were lost completely and the remaining 5,128,830 ha (57%) were impacted by varying degrees. For the 57% of the disturbed landscape, the average percent of these watersheds disturbed by human activity was calculated and was 28.25%. Severity Index = (0.43 x 100) + (0.57 x 28.25) = 59% resulting in a classification of Endangered since this value is above 50% and over 80% of its range was impacted.
Clarify severity for RLE D1. In calculating the average percent degraded of the not-completely-degraded watersheds, did you consider non-normality? At least it would be good to know as averages can be quite non-representative in some cases (in which case, you could use median perhaps).
Reply - We ran the same analysis using means and medians for the biological factors and the classified end results did not change.
Add simple description of methods for birds. Appendix S1 is too technical for a general audience. I’ve analysed bird data, and I have trouble understanding it (e.g., how is a PObs raster a species?). There’s no reason that the body of the paper can’t include a paragraph that describes methods simply. I question using species richness—some species listed are much more sensitive than others; for example, BCCH are common garden species, RBSA are associated with deciduous in some areas, while BRCR are well-established old growth specialists and NOGO need 60% foraging area to be at least mature. Some species listed do reasonably well with levels of retention even as low as those in policy. It’s difficult for me to review this section (even though I’ve published bird studies) because I don’t understand it. I’m guessing if I have trouble, others will have more difficulty.
Reply – to improve understanding of our bird analysis methods, we added lines 221-233. As to using species richness –species chosen for our analysis were associated with old growth based on the Atlas. It’s true that some maybe more than others but we did pull out the goshawk as an example, line 476. All studies that use measures of richness are prone to this problem and it comes down to the difference between choosing a synecology (community/guild) vs autecological approach. Assessing each species individually was beyond the scope of our study especially given we were dealing with multiple taxa to test the RLE biotic criterion. We added this concern to the limitations section lines 538-542. It is our hope that researchers will benefit by our experience in designing studies that specifically address RLE testing a priori and to avoid these pitfalls in the experimental design phase.
Increase discussion of RLE D1 relative to natural expected habitat. Caribou, old growth birds and specialised epiphytic lichens need old primary forest, not just primary forest. Thus, the comparison should ideally consider age as well as origin. It would be possible to use natural disturbance estimates to estimate the amount of old forest expected and use this in a sensitivity analysis. The discussion simply states that comparing to natural amounts was out of scope—I think a more fulsome discussion, including the ramifications of your assumptions (for example, that the proportion of old is the same within the remaining area as within the total managed plus primary area) is needed here.
Reply – while we agree that these biotic factors require old primary forest, we do not believe this would change our findings given managed areas are clearcuts-young plantations that are strikingly different from old primary forest. In terms of natural disturbance, we used the available fire perimeter data for the region that indicated ~10% has been impacted by fires caused by lightning, so a lot of our study area was very old forest before logging. We do not see the need to redo analyses at this stage.
Discuss assumption of persistence = remaining primary core forest. This is a big assumption and needs discussion. I understand that you’re looking for a way to mesh the data with the IUCN thresholds, but you should at least discuss other risk measures (e.g., sigmoidal 30-70% curves). Are there no ITR areas large enough in protected areas? I would say that uncertainties aren’t just due to unpredictable logging rates but also to the linear assumption.
Compare various approaches to determining ecosystem status (in the discussion likely). The introduction (para 1, line 42-47) mentions other assessment methods and concludes that they are “not comprehensive enough” for the study area. It doesn’t, however, present evidence of this lack (either here or in the discussion, unless I missed it), beyond noting they’re not identified as hotspots or key biodiversity areas yet. Given the claim (line 85) that this is the first application of RLE criteria for forests in North America, I would like some discussion of a) why other approaches wouldn’t work and b) how well you felt RLE applied. What are the costs and benefits? Would multiple pathways of evidence be useful? Or is this sufficient?
Reply – we agree with this addition and added lines 60-63 to the intro and lines 529-531 to the discussion. Interestingly, despite different methods and datasets, our rankings fell within the two WWF ecoregion status rankings – vulnerable to critical As how well the RLE performed, we already discussed uncertainties related to RLE application on lines 532-549.
Add climate change. The discussion would benefit from a climate change paragraph. Many of these species have had vulnerability assessed in a BC Fish and Wildlife Vulnerability database. https://www2.gov.bc.ca/assets/gov/environment/natural-resource-stewardship/nrs-climate-change/adaptation/climate20change20vulnerability20of20bcs20fish20and20wildlife20final20june6.pdf (I could only find the report, not the database, online).
Reply – good point – we added lines 523-527 to address climate vulnerabilities including the BC citation you referenced. Thank you.
Line 231. Did you include Old Growth Management Areas in the conservation status? In this area, there are non-legal and legal OGMAs. Although they offer limited protection, they’re worth mentioning particularly as they’re BC’s way of managing old forest.
Table 3. “missing data” seems preferable to “lack of information”. I think filling in the missing cells would be useful to see how much data are missing.
Reply - We added “missing dates” to Table 3 (lines 378-379). We are unsure what you mean by filling in missing cells. We used all the cut block data in the anthropogenic layer but for the logging by decade rate, we could not use cut block data because those data had no logging origin dates.
Line 293. It’s curious that the 1,419 watersheds averaged 4,443 ha—the identical mean size as the 3,722 watersheds mentioned in Table 4 (labelled 3) in section 4.3. Was this the mean size of the karst watersheds or of the total watersheds? If you know the number of karst watersheds, you can calculate the actual area and average size.
Reply – see lines 382-387 - Karst was assessed by focusing on watersheds with >10% of this feature to determine spatial extent for the RLE C1 thresholds. There were 1,410 watersheds totaling approximately 6.26 million ha (mean area x total number of watersheds with karst), which was around 38% of the study area. Of these, 1,154 (82%) were impacted by anthropogenic disturbances and the average percent human disturbed was 38%, which resulted in an overall REC C1 ranking of vulnerable [12].
Line 299 and Table “4”. Reference level is given at 10.2 million ha for IWB, yet Table 2 and section 2.2 say 9.4 million ha. Am I missing something here? If there’s a reason for the discrepancy, please explain.
Reply - The IWB study area was defined by intersecting the IWB data with the watershed file. The entire study area was 16.46M ha of which 9.4M ha were classified as actual IWB forest. Another 800,000 ha was classified as other forest types that comprised portions of the watersheds (primarily along the border of the study area boundary) when combined with the 9.4M ha resulted in a total area of 10.2M ha. The remaining 6.26M ha was high elevation rock & ice plus water bodies. This was all previously noted in our manuscript.
Figure 4. Consider grey scale for printing.
Reply – all our figures are in color. We are not sure why this particular figure should be switched to grey scale and would prefer to leave it as is for consistency with the other figures in color.
Minor edits
Why are endnotes roman numerals? They’re really hard to read.
Reply – looks like the editors took care of that, thankfully.
Section 1.
Line 62. I question whether primary forests are a province-wide priority in BC. The 2 cited references are both unpublished (there is a published version of reference #13 that should replace that one). While BC has recognised the importance of old growth, they have not recognised the importance of naturally-disturbed primary forests and still use age criteria to define old growth. Provincial government legislation, regulations and policy do not discuss primary forest. This paragraph seems to mix a couple of concepts (BC and Canada’s conservation priorities and the importance of IWB/ITR forest). I’d make it into two paragraphs. It’ll be important to have the comparison of primary and old growth forest (as described in the “clarify terminology” substantive suggestion above) before (or within) this section.
Reply - done – however, our study was not designed to distinguish primary from old growth. We are using primary to represent unlogged areas of all age classes, and in the case of the ITR, old growth is the predominant primary forest condition given extremely long fire rotation intervals. See lines 193-194.
Section 2.3
Line 133. Western redcedars are not “antique forests”; they are trees. Add a sentence on antique forests to the one on ancient trees.
Done
Section 2.4
Line 156-159. Perhaps listing the months for the quarters with stats would help. (e.g., add JJA after warmest quarter if those are the months included).
Done
Line 164. Awkward sentence. Fix.
Done - We split the sentence up
Section 2.5
Line 177. Typo. “respectfully” should be “respectively”.
Done
Line 186. “moisture-seeking plants, mosses, and hepatics (bryophytes)” is puzzling. All are plants. Mosses are also bryophytes. Clarify.
Done
Line 191. Typo: “form” should be “from”.
Done
Section 3.1
Line 202. Write out RLE in full the first time in the methods. I’d forgotten what it was by that point.
Done
Line 202 and on. These methods are ok, but you could use some friendlier language at least in the first sentence. “the product of a union among…” is unclear. Is “product” a mathematical operation (I think not). Union is correct, but maybe say “we overlaid three forest harvesting datasets…” in the first sentence and keep “union” for the explaining sentence afterwards. Also maybe explain that the three datasets are needed for the most accurate estimate of cutblocks.
Done
Line 209 “thru”??? Canadian spellings for Canadian ecosystems please!
Reply - Can you clarify please? We’re not sure what spellings you want changed.
Section 3.2
Line 234. Cool that you uploaded data (I haven’t had a chance to check it).
Table 1. The formatting on this table sucks on the version I have. It’s really challenging to understand. I’d also write out the rankings in full for ease of reading. The “extent and relative severity” bits seem strange—they don’t seem to differ among columns. A footnote would be better.
Reply –thanks for pointing that out – our challenge is to show all the rankings in one table considering the subcriteria/criteria differ in what is evaluated. We did our best to place them all in one table modeling the table after the original source – Bland et al. 2017. We think this clears up the table confusion.
Table 1. Criterion B1. Is the polygon within the study area or globally? Obviously, boundaries will dictate size. In this case, the ecosystem extends south of the Canadian border.
Reply – B1 was dropped as noted
Section 4.1
Line 247. “,” not “;”
Reply – we could not find this punctuation on line 247
Table 2. Lose the decimals from the %. Just makes it harder to read. I wonder if there’s a way to make the tables clearer?
Done
Table 2. Include the thresholds in the table title or in the table for ease (or at least refer folks to Table 1).
Added to Table descriptor.
Line 261-3. Update Table # references. This paragraph is important. I’d like the writing tightened and maybe the ranks written out. Even a minor change would help “Loss of core forest areas from anthropogenic disturbances was greater at 70% (IWB) and 95% (ITR), increasing the RLE rankings from none to endangered, and vulnerable to critical, respectively.”
Done
Section 4.2
Line 272. Is cutblock size relevant to the criterion? The criterion seems to focus on total amount (which is consistent with patterns of fragmentation). I’d suggest that instead of noting that cutblocks were 2.5x larger 1930s-60s you focus on the % of forest area logged as xx times more 1970s-2010s.
Reply – we added the decadal percentages but would prefer to leave in cut block sizes as is indicative of fragmentation effects as noted.
Table 3. “missing data” seems preferable to “lack of information”. I think filling in the missing cells would be useful to see how much data are missing.
Reply – noted “missing” – but not sure what you mean by filling in missing cells as we could not use the cut block data in the decadal rate estimates as noted because they had no cut block origin date.
Section 4.3
Table 4 mislabelled as Table 3.
Done
See RLE B1 substantive issue above. Suggest removing this section.
Done
Section 4.4
Line 296. “REC” should be “RLE” I assume. Please read for typos.
Done
Section 4.5
Line 299 and Table “4”. Reference level is given at 10.2 million ha for IWB, yet Table 2 and section 2.2 say 9.4 million ha. Am I missing something here? If there’s a reason for the discrepancy, please explain.
Table “4” is actually “5”. Check table #s throughout.
Reply - We doubled checked table numbering.
Figure 4. Caribou “core” habitat. Best use a different term as this could be confused with “core” (i.e., non-edge) forest.
Reply – caribou core is specific to the Canadian classification system for caribou habitat. We distinguish it from primary forest core areas. Therefore, we would prefer not changing this terminology.
Section 5.
Improve formatting of Table 5 (should be 6).
Reply – there are 5 tables, we double checked.
Line 374. Use Canadian designations rather than US.
Reply – unclear what the reviewer is wanting here, please clarify.
Line 381. True that salvage of pine is down, but demand is spruce everywhere, not just IWB. And definitely upslope in ESSF interior.
Reply – added upslope
Line 385. Ref 13 is now published.
Reply – added – thanks for that – good to see it published now – really important study!

Reviewer 2 Report
This is an interesting assessment of a rare ecosystem using the relatively new IUCN Red List of Ecosystems methodology. Note that this is not a truly scientific piece of research – it is not about hypothesis testing – but “simply” the application of guidelines to categorise the object of study. That being said, it is useful to publish and share given that the application of the guidelines has its challenges. My recommendation is to publish after the authors consider the comments provided by reviewers (i.e., minor corrections).
Going from theory to practice is not always an easy step, as shown by this piece of work. What looks like a simple task – applying the IUCN methodology to evaluate ecosystems against its Red List criteria – is in fact a tough undertaking. The authors do a good work in highlighting their assumptions and how they have overcome the difficulties of the task at hand. Also they present clearly what the impact of human activities over time has been on the two ecosystems that are the focus of this work. The paper is clearly written, and the conclusions drawn are well supported. There seems to be a bit of a misunderstanding about the Red List assessment itself, which is meant to assess the entire remaining ecosystem rather than bits of it, but in my opinion this is minor and does not affect the argumentation of the paper.
I don’t have major comments, but rather a series of more targeted remarks and suggestions for the authors to consider, which I paste below. Hope they are helpful.
Line 42: … is vital to forestall further loss [ii]….
I find the adjective vital a bit strong - a lot of things can be done without knowing all the specific details. Important or useful or helpful?
Lines 50-52: The IWB has not been identified as a hotspot or Key Biodiversity Area to date (although efforts to identify the latter in Canada are at an early stage).
This is likely to be the result of the lack of range restricted species or unique assemblies of vertebrate species. It may change with time and additional data, but currently the KBA identification process is a bit biased towards areas that are rich in (endemic) threatened (vertebrate) species. The most recent guidelines provided by IUCN, which took several years to produce, are an effort to counterbalance this. We will see if it works in due time.
Lines 130-132: Contemporary forest structure and composition within the ITR developed some 6,000 to 2,000 years BP following retreat of Pleistocene glaciers [xxiv].
This suggest that rather than conserving the forest per se, the most important would be to ensure that the local natural dynamics that led to the formation of such unique forests can follow their due course – not trivial because, as you mention later in the paper, these are very slow (thousands of years), and we have no idea about how to conserve natural systems for much longer than a century (Yellowstone, the oldest park set up for nature conservation purposes, dates from 1872)
Lines 172-173: The ITR is perhaps the most species-rich lichen (epiphytic and oceanic cyanolichens) 172 temperate rainforest in the world [16, 17]
You will need to speed-up the Red List assessment of lichens in order to accelerate the nomination of this area as a KBA ;-)
Line 177: … , respectfully [xxxi]. …
Did you mean respectively?
Lines 189-192: Bird species …. Using data and distribution models …
Is there convergence between the data and the distribution models? Often the models tend to overestimate the true distribution of species, which are often influenced not just by physical factors but by historical ones that are harder to model.
Lines 265-267: Figure 3. Potential primary forest (A…. Losses 266 were based on the total anthropogenic disturbance layer.
As written in my previous comment, models sometimes overestimate true extent. Later in the paper (e.g., 298-300) you set the reference using existing data – namely the logging concessions – to estimate the original extent. I assume there is good convergence between the data from the logging concessions and that of the modelling? If yes, stating this somewhere could strengthen your case even more. If not, then it is not very clear why the result of the modelling is presented – except perhaps that it is easier to map?
Lines 355-358: Table 5.
Interesting as it is, this table is also rather counterintuitive for me. The Red List as used by IUCN is meant to assess the entire ecosystem (or the entire species). Of course it can also be used to evaluate subregions (or subspecies or populations), but in the context of your paper you are making the assessment of a full ecosystem. So I found the distinction between the ecosystems and their core areas somewhat confusing. I understand your reasoning – i.e., highlighting that the ITR in particular is actually more threatened than the assessment of the full ecosystem indicates. But this is cheating the system, isn’t it? Or perhaps it is important to highlight point in your discussion: applying robotically the RL criteria will result in a serious underestimate of the risk for these ecosystems. And then again, the RL does not require that all criteria score equally (often there are only data to score one or two criteria), and it will apply the most threatened category found under any criterion as long as it is well supported – in this case EN-CR for the Interior Wetbelt (Criterion E) and EN for Inland Temperate Rainforest (B1), or CR if old-growth lichens apply equally to both (D1).
Lines 366-376: Criterion D1
For me one salient issue of your analysis is the fact that there is little overlap between some of these biotic factors (particularly Old growth birds), and the consequences that this entails. Any idea why this lack of concordance? Logging seems to be important to all, and presumably forest fragmentation has also influence upon old growth birds. In any case, your result highlight the difficulties of management the area – perhaps a gap analysis is needed to ensure all key elements are secured in a network - and the added risk for the full conservation of the ecosystem.
Lines 383-384: …that are prone to periodic insect outbreaks and salvage logging.
Are these insect outbreaks a result of the logging, or why is it relevant to mention them? It is not completely clear from the text.
Lines 406-407: Additionally, spatial distribution criteria in A1 and B1 were identical …
But that is not necessarily a problem with the RLE. Instead, the fact that the ecosystem you are focusing on has a restricted distribution results in this convergence between A1 and B1.
Lines 408-409: Assessing Criteria 408 D1 proved challenging…
It is! Perhaps the most challenging criteria of all, so really interesting to see how you tackled it. The RLE is a relatively new tool that will continue being refined and further developed.
Line 410: Additionally, we used remaining core, not potential core forest…
The RL uses the actual situation rather than the potential one whenever possible.
Lines 413-414: it is reasonable to assume that for at least the ITR most of it was primary forest prior to industrial logging.
True - a reasonable assumption. But this also indicates that its distribution has always been restricted, and even without the existing human pressures might end up as Endangered or Vulnerable under Criterion B.
Lines 422-424: The critical need for rapid action is revealed by the fact that the core functions of the ITR are within a decade or two of collapse.
The core area is what seems to be at grave risk of collapsing. What are the specific functions of the ITR that you are referrin
Author Response
Reviewer 2 – we appreciate the suggested edits and have made or responded to all of them as follows.
Going from theory to practice is not always an easy step, as shown by this piece of work. What looks like a simple task – _applying the IUCN methodology to evaluate ecosystems against its Red List criteria – _is in fact a tough undertaking. The authors do a good work in highlighting their assumptions and how they have overcome the difficulties of the task at hand. Also they present clearly what the impact of human activities over time has been on the two ecosystems that are the focus of this work. The paper is clearly written, and the conclusions drawn are well supported. There seems to be a bit of a misunderstanding about the Red List assessment itself, which is meant to assess the entire remaining ecosystem rather than bits of it, but in my opinion this is minor and does not affect the argumentation of the paper.
Reply – we appreciate your acknowledgement of our approach and the difficulties inherent to applying the RLE criteria. We indeed offer a case study of RLE application with its contributions and limitations as stated in our discussion section. Thank you
L_i_n_e_ _4_2_:_ _… _i_s_ _v_i_t_a_l_ _t_o_ _f_o_r_e_s_t_a_l_l_ _f_u_r_t_h_e_r_ _l_o_s_s_ _[_i_i_]_…._ _
I find the adjective vital a bit strong - a lot of things can be done without knowing all the specific details. Important or useful or helpful?
Reply – given the extent and speed of deforestation and forest degradation globally, we would prefer not to soften the wording here. We refer the reviewer to the Global Forest Watch forest tracking site for rate of forest loss globally and regionally -
https://www.globalforestwatch.org/. To us, such losses are indeed alarming and should not be understated given the biodiversity and climate emergencies (see papers by Ripple et al. 2017, 2020 regarding climate emergency and biodiversity losses).
Lines 50-52: The IWB has not been identified as a hotspot or Key Biodiversity Area to date (although efforts to identify the latter in Canada are at an early stage).
This is likely to be the result of the lack of range restricted species or unique assemblies of vertebrate species. It may change with time and additional data, but currently the KBA identification process is a bit biased towards areas that are rich in (endemic) threatened (vertebrate) species. The most recent guidelines provided by IUCN, which took several years to produce, are an effort to counterbalance this. We will see if it works in due time.
Reply – yes – we agree – that is definitely the case given the taxa bias and with the high lichen richness/endemicity there is certainly cause for KBA evaluations and we added that to line 563-565.
Lines 130-132: Contemporary forest structure and composition within the ITR developed some 6,000 to 2,000 years BP following retreat of Pleistocene glaciers
This suggest that rather than conserving the forest per se, the most important would be to ensure that the local natural dynamics that led to the formation of such unique forests can follow their due course – _not trivial because, as you mention later in the paper, these are very slow (thousands of years), and we have no idea about how to conserve natural systems for much longer than a century (Yellowstone, the oldest park set up for nature conservation purposes, dates from 1872)
Reply – the local natural dynamics are clearly important and they are changing. The best we can do at this point is to stop the loss of primary/old-growth forests as noted in several places in our paper.
Lines 172-173: The ITR is perhaps the most species-rich lichen (epiphytic and oceanic cyanolichens) 172 temperate rainforest in the world [16, 17]
You will need to speed-up the Red List assessment of lichens in order to accelerate the nomination of this area as a KBA ;-)
Reply – thanks for this comment – we added that to lines 5663-565.
L_i_n_e_ _1_7_7_:_ _… _,_ _r_e_s_p_e_c_t_f_u_l_l_y_ _[_x_x_x_i_]_._ _… _
Reply - Good catch – we corrected this.
Lines 189-1_9_2_:_ _B_i_r_d_ _s_p_e_c_i_e_s_ _…._ _U_s_i_n_g_ _d_a_t_a_ _a_n_d_ _d_i_s_t_r_i_b_u_t_i_o_n_ _m_o_d_e_l_s_ _… _
Is there convergence between the data and the distribution models? Often the models tend to overestimate the true distribution of species, which are often influenced not just by physical factors but by historical ones that are harder to model.
Reply - It’s true that models do not converge for all species, due to any number of reasons (these are often difficult to unravel). However, that is not the case for the species used in this paper, as they all had good models. In total, 240 of 351 species from the British Columbia Breeding Bird Atlas (the source used here) had reasonable distribution models that were reviewed extensively by experts who checked for exaggerations in range extent. With extremes in elevation, moisture, temperature, etc., present in BC, the model review and refinement process were particularly rigorous and careful.
Lines 265-2_6_7_:_ _F_i_g_u_r_e_ _3_._ _P_o_t_e_n_t_i_a_l_ _p_r_i_m_a_r_y_ _f_o_r_e_s_t_ _(_A_…._ _L_o_s_s_e_s_ _2_6_6_ _w_e_r_e_ _b_a_s_e_d_ _o_n_ _t_h_e_ _t_o_t_a_l_ _anthropogenic disturbance layer.
As written in my previous comment, models sometimes overestimate true extent. Later in the paper (e.g., 298-300) you set the reference using existing data – namely the logging concessions – to estimate the original extent. I assume there is good convergence between the data from the logging concessions and that of the modelling? If yes, stating this somewhere could strengthen your case even more. If not, then it is not very clear why the result of the modelling is presented – except perhaps that it is easier to map?
Reply – actually, we did not do any modeling for this part of the paper. The anthropogenic layer was a simple mapping of all the human disturbances. The cutblock layer was a mapping of the 3 cut block datasources we were able to obtain from the BC government. No modeling was done in this analysis.
Lines 355-358: Table 5.
Interesting as it is, this table is also rather counterintuitive for me. The Red List as used by IUCN is meant to assess the entire ecosystem (or the entire species). Of course it can also be used to evaluate subregions (or subspecies or populations), but in the context of your paper you are making the assessment of a full ecosystem. So I found the distinction between the ecosystems and their core areas somewhat confusing. I understand your reasoning – _i.e., highlighting that the ITR in particular is actually more threatened than the assessment of the full ecosystem indicates. But this is cheating the s_y_s_t_e_m_,_ _i_s_n_’t_ _i_t_?_ _O_r_ _p_e_r_h_a_p_s_ _i_t_ _i_s_ _i_m_p_o_r_t_a_n_t_ _t_o_ _h_i_g_h_l_i_g_h_t_ _p_o_i_n_t_ _i_n_ _y_o_u_r_ _d_i_s_c_u_s_s_i_o_n_:_ _a_p_p_l_y_i_n_g_ _r_o_b_o_t_i_c_a_l_l_y_ _the RL criteria will result in a serious underestimate of the risk for these ecosystems. And then again, the RL does not require that all criteria score equally (often there are only data to score one or two criteria), and it will apply the most threatened category found under any criterion as long as it is well supported – _in this case EN-CR for the Interior Wetbelt (Criterion E) and EN for Inland Temperate Rainforest (B1), or CR if old-growth lichens apply equally to both (D1).
Reply – actually, we think our approach was robust to this concern because it was a multi-scaled analysis: bioregion (IWB), subzone (ITR), watersheds, core areas; the rankings supported one another across scales. So, we do not believe we cheated the system or the ranking approach in any way, given that all three scales combine to tell a robust story of imminent collapse, particularly when you examine the core functions of the ecosystem compromised by cumulative fragmentation. Had we not done it this way, we would have grossly underestimated losses.
Lines 366-376: Criterion D1
For me one salient issue of your analysis is the fact that there is little overlap between some of these biotic factors (particularly Old growth birds), and the consequences that this entails. Any idea why this lack of concordance? Logging seems to be important to all, and presumably forest fragmentation has also influence upon old growth birds. In any case, your result highlight the difficulties of management the area – perhaps a gap analysis is needed to ensure all key elements are secured in a network - and the added risk for the full conservation of the ecosystem.
Reply – yeah, good point. Perhaps what is going on here is related to extent of habitat available. The lichens depend on the wettest ITR and that has already been reduced substantially particularly core conditions. OG birds are not entirely restricted to the ITR and can use old forests in the IWB as well. So, their range is more expansive. It comes down to what was available previously vs. what remains currently we suppose. Craighead and Cross (cited in our paper) did a GAP analysis on this region using multiple taxa and found 45% of the area needed some form of protection in a conservation area design, which is nearly double the amount protected currently.
Lines 383-3_8_4_:_ _…t_h_a_t_ _a_r_e_ _p_r_o_n_e_ _t_o_ _p_e_r_i_o_d_i_c_ _i_n_s_e_c_t_ _o_u_t_b_r_e_a_k_s_ _a_n_d_ _s_a_l_v_a_g_e_ _l_o_g_g_i_n_g_._ _
Are these insect outbreaks a result of the logging, or why is it relevant to mention them? It is not completely clear from the text.
Reply – to our knowledge there hasn’t been any work linking epizootic outbreaks to logging practices directly – most likely it’s a climate signal, particularly the upper elevation areas where spruce outbreaks are occurring. Line 490 does show that it’s “subsequent post-disturbance logging” – meaning the logging follows the outbreak and not vice-versa.
Lines 406-407: Additionally, spatial d_i_s_t_r_i_b_u_t_i_o_n_ _c_r_i_t_e_r_i_a_ _i_n_ _A_1_ _a_n_d_ _B_1_ _w_e_r_e_ _i_d_e_n_t_i_c_a_l_ _… _
But that is not necessarily a problem with the RLE. Instead, the fact that the ecosystem you are focusing on has a restricted distribution results in this convergence between A1 and B1.
Reply – we decided to drop this criterion from analysis (based on advice from Reviewer 3) and we pointed to problems with spatial overlap in the criteria A vs B as applied to our region. See lines 532-535.
Lines 408-409:_ _A_s_s_e_s_s_i_n_g_ _C_r_i_t_e_r_i_a_ _4_0_8_ _D_1_ _p_r_o_v_e_d_ _c_h_a_l_l_e_n_g_i_n_g_… _
It is! Perhaps the most challenging criteria of all, so really interesting to see how you tackled it. The RLE is a relatively new tool that will continue being refined and further developed.
Reply – thanks! Appreciate that – and we continue to hope that our case study application will result in adaptive learning for the RLE process.
Line 410: Add_i_t_i_o_n_a_l_l_y_,_ _w_e_ _u_s_e_d_ _r_e_m_a_i_n_i_n_g_ _c_o_r_e_,_ _n_o_t_ _p_o_t_e_n_t_i_a_l_ _c_o_r_e_ _f_o_r_e_s_t_… _
The RL uses the actual situation rather than the potential one whenever possible.
Lines
Reply – good to know that we did this consistent with the intent of the RLE. Thanks!
Lines 413-414: it is reasonable to assume that for at least the ITR most of it was primary forest prior to industrial logging.
True - a reasonable assumption. But this also indicates that its distribution has always been restricted, and even without the existing human pressures might end up as Endangered or Vulnerable under Criterion B.
Reply – that is certainly the case but we dropped B1 based on Reviewer3 concerns.
Lines 422-424: The critical need for rapid action is revealed by the fact that the core functions of the ITR are within a decade or two of collapse.
The core area is what seems to be at grave risk of collapsing. What are the specific functions of the ITR that you are referring
Reply – our study was not designed to test core functions impacted by edge effects. We used a conservative 100-m buffer as a proxy for edge effects reported in the literature; effects are many – biotic (e.g., nest predation) and abiotic (e.g., dessication, wind, temperature differences). We added this to lines 471-473.

Reviewer 3 Report
The study analyzes the overall conditions of the interior wetbelt and inland temperate forests in Canada with IUCN Red List Ecosystem Categories and Criteria. The study is interesting and fitting for publication in the journal Land.
However, in my opinion, the authors need to make some minor changes in the manuscript before it is published. Please see my detailed comments on these suggested changes below:
Section 1: I feel that the authors can expand the usability part of the RLE criteria. Some real world examples of how RLE criteria is used for protection/conservation of ecosystems will be very much in line with the applicability of the study (see also the usability of the study below).
How concepts and methodologies connect to the real ecosystems (and for conservation of landscapes) is an important aspect of nature conservation, which this study tries to connect. This aspect can be brought to the forefront in a more clearcut manner.
Section 2.4: The first paragraph of the section needs to be streamlined especially with the study. The abiotic geologic and geomorphic structure and processes need to be linked with the forest ecosystem, the way the authors have linked the climatic factors.
For example, any special forest structure/characteristics for the different geologic structures (i.e., sedimentary and metamorphic Cariboo Range; gneissic metamorphic, sedimentary volcanic, intrusive igneous Monashee Range; sedimentary, metamorphic, volcanic and intrusive igneous Selkirk Range, sedimentary and metamorphic Purcell Range)?
In the subsections forest types, structure, abiotic and biotic factors related sections should cross refer each other whenever possible to increase the streamlining.
Section 2.5 needs to give some basic but important information (mention in-text) about mammals and other species of animals (at present only caribou is mentioned), that play a pivotal role to forest ecosystem (for example the predators). Also it will be better to mention in-text some rare and endangered species names to attract more conservation attention.
Methods: The methods section needs to be written in an integrated way. It needs to state clearly what are being measured, how they are being measured , and to find what in a better way. To do this the authors need to integrate the three questions they deal with in the last paragraph of the introduction section; i.e.,
"This analysis required an assessment of what remains, where it occurs, and how fast it is being depleted." This part needs to be clearly mentioned again in the methods to show us which part/s of the methods answer which question/s.
Line 294-295: "Of these, 854 karst watersheds with an estimated area of 3.7 million ha were impacted by anthropogenic disturbances,..." >> what are these anthropogenic disturbances? please mention.
Usability of the study: The study in my opinion, needs to put a short section/paragraph on the implications of the study for the future/possible protection/ conservation of the ecosystems the authors deal with in the paper. For example, what the BC and Canadian government needs to do to conserve these forests? Where can the conservation pathways start based on the authors experience and the results of this study?
Conclusion: The conclusion section can be improved. Pleas estate again the main findings of the research; mentioning the novelty of this study. The usability of the study (see above) can be summerized in a sentence or two in the conclusion.
Citation style: Please follow standard citation style of the journal. The in-text citation should be in Arabic numbers (not Roman numbers).
Author Response
Reply to Reviewer 3– we thank the reviewer for thoughtful suggestions and we have done our best at incorporating most of them while explaining places where we believe the suggestions would detract from the main purpose of the study. We wish to reiterate – one of the main reasons for this study was to test the RLE criteria for a region that has not been examined previously. So, we need to focus on recommendations that improve our ability to apply the RLE criteria, which is also the focus of this special feature in Land.
The study analyzes the overall conditions of the interior wetbelt and inland temperate forests in Canada with IUCN Red List Ecosystem Categories and Criteria. The study is interesting and fitting for publication in the journal Land.
However, in my opinion, the authors need to make some minor changes in the manuscript before it is published. Please see my detailed comments on these suggested changes below:
Section 1: I feel that the authors can expand the usability part of the RLE criteria. Some real world examples of how RLE criteria is used for protection/conservation of ecosystems will be very much in line with the applicability of the study (see also the usability of the study below).
Reply – we added lines 65-76 on applicability of RLE for other forest ecosystems. However, please note, our understanding is that the special feature in Land is designed to cover examples of the RLE approach. While we mention that our study is a case study of RLE application, we were not sure if the editors would like us to mention the special feature in our intro or if it’s obvious considering the papers are being grouped together in this special feature. More guidance would be appreciated so we know whether to cite the special feature or not. Regardless, we are not privy to all the articles thus far received.
How concepts and methodologies connect to the real ecosystems (and for conservation of landscapes) is an important aspect of nature conservation, which this study tries to connect. This aspect can be brought to the forefront in a more clearcut manner.
Reply – we agree and have added lines 65-92. Thank you for this suggestion.
Section 2.4: The first paragraph of the section needs to be streamlined especially with the study. The abiotic geologic and geomorphic structure and processes need to be linked with the forest ecosystem, the way the authors have linked the climatic factors.
For example, any special forest structure/characteristics for the different geologic structures (i.e., sedimentary and metamorphic Cariboo Range; gneissic metamorphic, sedimentary volcanic, intrusive igneous Monashee Range; sedimentary, metamorphic, volcanic and intrusive igneous Selkirk Range, sedimentary and metamorphic Purcell Range)?
In the subsections forest types, structure, abiotic and biotic factors related sections should cross refer each other whenever possible to increase the streamlining.
Reply –the only abiotic factor we examined was karst because of its association with rare plants as mentioned on lines 173-74 and 538. We did not examine any other abiotic or geological feature so linking to forest structure is not needed. We believe the information on mountain ranges is for context that most studies provide based on the physical features of the study area and we would like to retain this as descriptive (context). Additionally, the RLE abiotic criteria as published by Bland et al. 2017 include contextual information and thus we are following the protocol of providing at least some geological features to remain as close as possible to the task at hand – applying the RLE approach to our study area.
Section 2.5 needs to give some basic but important information (mention in-text) about mammals and other species of animals (at present only caribou is mentioned), that play a pivotal role to forest ecosystem (for example the predators). Also it will be better to mention in-text some rare and endangered species names to attract more conservation attention.
Reply –we chose one mammal focal species (caribou) to examine and linked that focal species to other biotic criteria evaluated such as lichens (line 207). We prefer to stick to species that fit the RLE criteria as applied to our study. Adding more species would detract from the focal species approach that followed the biotic RLE criteria. Again, we were limited by what we could analyze to test specific criteria and while we agree there is more to this story, we hope you agree that sticking to what we could analyze to test the criteria is the best approach at this time.
Methods: The methods section needs to be written in an integrated way. It needs to state clearly what are being measured, how they are being measured, and to find what in a better way. To do this the authors need to integrate the three questions they deal with in the last paragraph of the introduction section; i.e.,
"This analysis required an assessment of what remains, where it occurs, and how fast it is being depleted." This part needs to be clearly mentioned again in the methods to show us which part/s of the methods answer which question/s.
Reply – thank you for this suggestion – we added line 241-243 and believe this addresses your concern.
Line 294-295: "Of these, 854 karst watersheds with an estimated area of 3.7 million ha were impacted by anthropogenic disturbances,..." >> what are these anthropogenic disturbances? please mention.
Reply – line 246 already states the impacts were anthropogenic disturbances, which in turn, were described in the preceding Section 4.1 (line 305 in particular). Thus, this suggestion seems redundant.
Usability of the study: The study in my opinion, needs to put a short section/paragraph on the implications of the study for the future/possible protection/ conservation of the ecosystems the authors deal with in the paper. For example, what the BC and Canadian government needs to do to conserve these forests? Where can the conservation pathways start based on the authors experience and the results of this study?
Conclusion: The conclusion section can be improved. Pleas estate again the main findings of the research; mentioning the novelty of this study. The usability of the study (see above) can be summerized in a sentence or two in the conclusion.
Reply – we added a bit more to the discussion (lines 529-549) and conclusions (lines 551-567) to highlight significance of our findings, thank you.
Citation style: Please follow standard citation style of the journal. The in-text citation should be in Arabic numbers (not Roman numbers).
Reply – looks like the editors, thankfully, handled that one in layout.
